

**The ENSO teleconnections to the Indian summer monsoon climate through the Last Millennium**
**as simulated by the PMIP3.**
**Charan Teja Tejavath[a], Karumuri Ashok[a], Supriyo Chakraborty[b] and Rengaswamy Ramesh[c]**
**Corresponding author: ashokkarumuri@uohyd.ac.in**
[a]Centre for Earth, Ocean and Atmospheric Sciences, University of Hyderabad, Hyderabad, India.
[b]Indian Institute of Tropical Meteorology, Pune, India.
[c]School of Earth and Planetary Sciences, NISER, Bhubaneswar, India.
**Abstract**
Using seven model simulations from the PMIP3, we study the mean summer (June-September)
climate and its variability in India during the Last Millennium (LM; CE 850-1849) with emphasis on the
Medieval Warm Period (MWP) and Little Ice Age (LIA), after validation of the simulated 'current day'
climate and trends.
We find that the above (below) LM-mean summer global temperatures during the MWP (LIA)
are associated with relatively higher (lower) number of concurrent El Niños as compared to La Niñas.
The models simulate higher (lower) Indian summer monsoon rainfall (ISMR) during the MWP (LIA).
This is notwithstanding a strong simulated negative correlation between the timeseries of NINO3.4
index and that of the area-averaged ISMR, Interestingly, the percentage of strong El Niños (La Niñas)
causing negative (positive) ISMR anomalies is higher in the LIA (MWP), a non-linearity that apparently
is important for causing higher ISMR in the MWP. Distribution of simulated boreal summer velocity
potential at 850 hPa during MWP in models, in general, shows a zone of anomalous convergence in the
central tropical Pacific flanked by two zones of divergence, suggesting a westward shift in the Walker
circulation as compared to the simulations for LM as well as and a majority of historical simulations,
and current day observed signal. The anomalous divergence centre in the west also extends into the
equatorial eastern Indian Ocean, resulting in an anomalous convergence zone over India and therefore
excess rainfall during the MWP as compared to the LM; the results are qualitative, given the inter-model
spread.



**Introduction**

3       Instrumental records of climate seldom date back prior to the 1850s. Therefore, analysis of

proxy climate data, aided by climate modelling, has been the principal means to evaluate past climate
variability. Past climate records exhibit significant variability on millennial to interannual time scales
(IPCC, 2013). Interestingly, this IPCC report based on a large number of publication points out
significant centennial climate variations during the last two millennia (PAGES 2k Consortium, 2013),
though there is apparently no significant anthropogenic influence similar to the second half of the 20th
century. Paleo-data based studies such as those by Lamb et al., (1965); Grove et al., (1988); Graham et
al., (2010) Mann et al., (2009) identify two significant periods in the last millennium (LM) prior to the
period when instrumental observations started, i.e.  Common Era (CE) 850-1849. These are, (i) a
relatively warmer period known in literature as the 'Medieval Warm Period' (MWP, CE 950-1350),
roughly followed by (ii) a relatively cooler period, the Little Ice Age (LIA, CE 1500-1850). The
presence of these warmer (MWP) and cooler (LIA) periods varies from region to region, in terms of
timing, duration and magnitude of the temperature anomalies.
Paleoclimate reconstructions from various well-dated proxy data suggest that during the MWP,
some regions experienced temperatures as warm as mid-20th century, whereas some others were as
warm as the late-20th century (e.g., IPCC 2013, Prasad and Enzel, 2006; Fleitmann et al., 2007; Ponton
et al., 2012).
The Indian Summer Monsoon Rainfall (ISMR; June-September; JJAS) variability is manifested
on intra-annual, interannual, decadal, centennial and millennial to multi-millennial time scales (Ramesh
et al., 2010). Paleo-monsoon records from well-dated proxy data from the Arabian Sea (e.g. Sarkar et
al., 2000; Gupta et al., 2003; Staubwasser et al., 2003; Tiwari et al., 2005), the Arabian Peninsula (e.g.
Fleitmann et al., 2007; Fleitmann et al., 2003; Neff et al., 2001), and the Indian sub-continent (e.g.
Berkelhammer et al., 2012; Dixit et al., 2015; Dixit et al., 2014a; Dixit et al., 2014b; Dixit, 2013; Dutt et
al., 2015; Nakamura et al., 2015) show centennial-to millennial-scale changes in the ISMR during the
Holocene.
In a recent review, Dixit and Tandon (2016) suggest that MWP and LIA effects are well
reflected in the ISMR, with a caveat that proxy data exhibit heterogeneity in terms of the timing and
duration. Proxy records also suggest that, by and large, during the last millennium, ISMR was the
highest during the MWP and relatively weaker during the LIA (Yadava et al., 2005). However, the data



density is rather sparse in time and space to quantify the decadal through the centennial scale temporal
structure of ISMR variability during MWP and LIA.
A speleothem-based reconstruction of ISMR variability by Sinha et al., (2007) exhibits an
evolution conforming to solar activity (for which atmospheric radiocarbon activity is used as a
surrogate) only during the MWP. An increased summer monsoon precipitation during the MWP is
suggested to be linked to the ENSO-modulated solar forcing in proxy studies by Berkelhammer et al.,
(2010) and Emile-Geay et al., (2007). The speleothem-based monsoon reconstruction of Sinha et al.,
(2007 and 2011) suggests a severe weakening of Indian Summer Monsoon (ISM) during the LIA,
apparently associated with multi-year to decades long  droughts particularly between 13[th] and 17[th]
centuries. Another proxy record, from the Dandak cave in Central India, shows a 30% rainfall reduction
during the 14 century (Yadava, et al., 2005).
Obviously, the recent ~150-year period is the best documented period in terms of instrumental
observations. Uncertainties, however, exist in terms of the quality and spatial density of data even for
this period.
The observational records of ISMR from the beginning of the last century show that its
interannual and inter-decadal variability is significantly associated with that of the El Niño-Southern
Oscillation (ENSO) (e. g. Keshavamurty 1982, Sikka 1980; see Ashok et al., 2004 for further
references). Typically, the warmer (cooler) ENSO events are associated with lesser (higher) than normal
rain over India during the boreal summer, concurrent with the Indian monsoon season. Prasad et al.
(2014) based on proxy climate data, infer that the long-term influence of ENSO like conditions on ISM
began only 2ky BP, and is coincident with Southern Indo-Pacific warm pool (IPWP) warming. They
also suggest that the IPWP-ISM links and large scale advection of moist air toward India varies on a
multi-centennial scale. Kitoh et al., (2007), in a model study, observed decadal variability in the ENSO-
ISM relation. Through a 31-yr moving correlation analysis, they show that, during the LM, monsoon-
ENSO correlations vary over a wide range, specifically -0.71 to +0.07, with an overall correlation of -
0.34 for the LM.
Thus, the variability of Indian summer monsoon during the LM has been relatively less studied,
particularly from the modelling perspective. It is also noticeable that all the model studies cited above
primarily employed *single* GCMs. From this perspective, it is interesting to explore multi-model
simulations such as those from the PMIP3, to study Indian summer monsoon conditions during the LM,

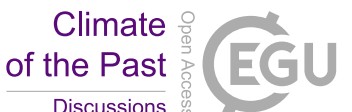

specifically the MWP and the LIA, and examine whether these model results could be reconciled with proxy-observations. Likewise, such a study highlights the capability of these models in capturing at least a millennium of the past climate with fidelity, in addition to facilitating a quantification of the multi-model spread. Furthermore, such a study would serve as a benchmark for addressing longer periods of climate variability relevant to the Indian summer monsoon using models.

With this motivation, here we study the multi-model simulated ISMR variability and its teleconnections with the ENSO during the LM, using various relevant PMIP3 datasets with an emphasis on the simulated Medieval Warm Period (MWP; CE 1000-1199) and Little Ice Age (LIA; CE 1550-1749). We consider the 200 warmest (relatively coldest) year-period as the simulated MWP (LIA) period for maintaining uniformity between global and regional analysis of ENSO-ISM teleconnections from CMIP5 LM simulations, with the knowledge that the temporal and spatial signatures of the MWP and LIA varied from region to region, at least in terms of magnitude (e.g. Dixit and Tandon 2016).

In the following sections, we describe the various reanalysed, observed, and PMIP3 datasets we used, present our results subsequently, and finally provide a concluding summary.

**Data and Methodology**

It is indeed a challenging prospect to validate the simulated Indian summer monsoon features from the PMIP3 simulations for the LM period given the sparse and scanty observations. Fortunately, the corresponding model simulations of the CMIP5 for the historical period (CE 1850-2005), i.e. the current day climate, can be validated using various observed/reanalysed gridded datasets, keeping in mind the uncertainties associated with such datasets during the pre-satellite period. Therefore, in this study, we start by exploring the fidelity of simulated Indian summer monsoon climate from historical simulations (henceforth referred to as HS) that cover the CE 1850-2005 period for which instrumental observations are available. It may be noted that this exercise is carried out only for seven CMIP5 models for which the PMIP3 simulations for the LM period are available for the CE 850-1849 period (LM), under the class termed as 'past1000 (henceforth referred to as p1000)'.

For the HS, the models were forced to use the observed atmospheric composition changes with natural aerosols or their precursors, and natural sources of short-lived species, and time-evolving land cover as outlined by Taylor et al. (2012). On the other hand, the p1000 results were obtained by forcing the models with well-mixed greenhouse gases, changes in volcanic aerosols, land use, and solar





irradiance changes (Taylor et al, 2012; Schmidt et al., 2011; Schmidt et al., 2012). We evaluate the
fidelity of the HS simulations by comparing with the observed/reanalysed Indian summer monsoon
rainfall and air temperature. The seven models whose data used in this study are: BCC-CSM-1-1(m),
IPSL-CM5A-LR, FGOALS-s2, MPI-ESM-P, GISS-E2-R, CCSM4 and HadCM3. These datasets have
been downloaded from "http://cera-www.dkrz.de/WDCC/ui/Index.jsp". The acronyms used and details
for these datasets are presented in Table 1. The various observational/reanalysed data sets used for the
validation of the HS are, the Hadley Centre Interpolated sea surface temperature (HadISST; Titchner
and Rayner, 2014) for CE 1870-2014, the ERA-20CM sea surface temperature and skin temperature
(SST and SKT respectively; Hersbach et al. 2015) available for CE 1900 to 2010 (using two sea surface
temperature datasets throws light on any uncertainties associated with the data quality therein) and the
India Meteorological Department (IMD) gridded rainfall datasets for CE 1901-2009 period, available
at 1.0º latitude x 1.0º longitude resolution and covering the land region bound by 66.5º E-101.5º E; 6.5º
N-39.5º N (Rajeevan et al., 2006). NCEP/NCAR Reanalysis 1 datasets of variables Eastward wind and
Northward wind (U-Wind & V-Wind) at pressure levels available from 1948 to present (Kalnay et al.,
1996) are also used. For uniformity, all the simulated precipitation and near air surface temperature data
sets were re-gridded to 2.0º latitude x 2.0º longitude resolution grids. The historical simulations from the
individual models are validated by comparing various climate statistics with the corresponding climate
statistics from observed and reanalysed datasets for the CE 1901-2005 period.
We use the well-known NINO3.4 index, an area-averaged SST anomaly over the region bound
by 170ºW-120ºW; 5ºS-5ºN to represent the ENSO variability. An Indian summer monsoon rainfall
(ISMR) index is obtained by area-averaging the mean June-through-September (JJAS) rainfall over the
land region bound by 65ºE-95ºE; 10ºN-30ºN. The area-averaged temperature for the Indian region is
also obtained by averaging the surface temperature over this region.
To check the ENSO-ISM relationship and its longterm variability during LM, we calculate the
monthly anomalies of surface temperature and precipitation from their respective climatological
monthly means. The anomalies of any parameter, such as, say, the JJAS temperature, for each model
have been obtained by subtracting the 1000-year climatological value of the individual seasonal values.
Linear correlation analysis is used to estimate the ENSO-ISMR relationship during various periods.
We have also explored the relevance of the simulated land-sea thermal gradient (LSTG)
between the Indian land temperatures during pre-monsoon (i.e. April-May), and that during summer
monsoon, for the ISMR (e.g. Pant and Kumar et al. 1997; Roxy et al. 2015). Given its importance, we



use two slightly different indices to represent the LSTG by considering two different land regions (RG1)
most of the Indian land region encompassed by 70°E-90°E, 5°N-35°N (e.g. Roxy et al., 2015), and
(RG2) a land region 65°E-80°E,25°N-35°N, which covers the northwest Indian sub-continent covering
Pakistan and the desert region of Indian subcontinent to its east, known to be very hot during pre-
monsoon months. The LSTG indices have been obtained by subtracting the area-averaged SST over
ocean region 50°E-65°E,5°S-10°N (Roxy et al., 2015) from the area-averaged temperature from the land
boxes mentioned above.
We carry out a trend analysis, the significance of which has been evaluated through the Mann-
Kendall test. The statistical significance of linear correlation, and that of the partial correlation, has been
evaluated using a 2-tailed Student's t-test. Further, while ascertaining the statistical significance of
correlation differences from MWP to LIA, we  employ a boot-strapping test as well.
**3 Results**
*3.1 Validation of the HS*
Figures A1a and A1b respectively show the 11-year running mean of near-surface air
temperatures globally-averaged, and averaged over Indian region, from the seven models of the HS;
Figures A1c and A1d show the corresponding time series of anomalies. It is seen from Figures A1c and
A1d that all the models can simulate the observed increasing temperature trend reasonably,
notwithstanding an inter-model spread. Further, we find that the observed as well as and the simulated
trends are significantly above the corresponding interannual standard deviations (e.g. Figure SPM.1a;
Figure TS. 1; Figure TS. 9; Stocker et al., 2013; IPCC, 2013;).  Figure A1d suggests that the surface
temperatures over India also have continued to rise till the end of 20[th] century, which agrees with
observations (Revadekar et al., 2012). Several recent studies suggest a decreasing trend in Indian
summer monsoon rainfall (e.g. Guhathakurtha et al., 2007; Krishnan et al., 2016; Sano et al., 2011) in
recent decades. Figure A2a and A2b show the inter-model spread across the models with the
corresponding observations. We find statistically decreasing trend in four models at the end of 20th
Century in agreement with the observations. The trends in the other models are not statistically
significant. We revise the text accordingly.
On a different note, an increase in warm ENSO events, be it canonical or Modoki (e.g. Ashok et
al., 2007), has been observed in the late 20th century with an increase in global temperature (e.g. Collin
2000; Cai et al. 2015). The models are able to reproduce this trend qualitatively to a reasonable extent,



as seen by the higher number of simulated warm events, represented by the positive Nino3.4 index
(Figure A2c).
That the ENSO is a major driver of interannual variability of the Indian summer climate is
evidenced by the negative correlation of -0.50 (Figure 1a) between the time series of ISMR and
NINO3.4 index derived from the HadISST for the period CE 1901-2005, statistically significant at 0.01
level from a 2-tailed Student's t-test. Note that the corresponding correlation obtained by using the
NINO3.4 index from the ECMWF SST data sets is -0.57. The corresponding NINO3.4-ISMR
correlations from the HS are also presented in Figure 1a. Five out of the seven models simulate the
negative correlations with a range of -0.21 to -0.51, which are statistically significant at 0.05 levels from
a 2-tailed Student's t-test. The CCSM4 and FGOALS-s2 models simulate  weaker correlation
coefficients of -0.12 (significant at 0.2 level)  and -0.10, respectively.
The Indian summer temperature for CE 1901-2005 yields a moderate correlation coefficient of
0.34 and 0.38 respectively, with the concurrent NINO3.4 index from HadISST and that from the
ECMWF SST datasets; both values are statistically significant at 0.05 level from a 2-tailed Student's t-
test. Corresponding correlations for seven (five) models are statistically significant at 0.1 (0.05) level
from a 2-tailed test, though they vary over a wide range of values varying from 0.13 to 0.74 (Figure 1b).
Considering results from Figure 1, we surmise that all these seven models reproduce the ENSO
relation to the JJAS temperature and/or rainfall in the Indian region qualitatively well. More
importantly, Figures A1 show that these models are also able to capture the long term trends in the
summer monsoon temperature and Figure A2 show taht four models (BCC, CCSM4, GISS and MPI) are
capturing the decreasing trend in agreement with the observations. The interannual standard deviation
for these two parameters from observations as well as from the individual model simulations are
presented in Table S1. We find that simulated standard deviations from various models fall within a
±20% range of observations.
In summary, the BCC-CSM-1-1(m), IPSL-CM5A-LR, MPI-ESM-P, GISS-E2-R, CCSM4,
HadCM3 and FGOALS-s2 models meet our criteria for their p1000 simulations to be used for further
analysis to understand the LM variability.
***3.2 p1000 analysis***



To ascertain that there is a reasonable agreement of variability among the LM simulations from
the models, we present in Table S2 the JJAS standard deviations ($\sigma$) of the simulated area-averaged
global rainfall, area-averaged surface temperature, and the NINO3.4 index for the whole period as well
as three overlapping 500-year sub-periods, namely, CE 850-1349, CE1100-1599, and CE 1350-1849.
The simulated statistics from the individual models fall within the $\pm1\sigma$ range of the corresponding
statistic (Table S2) in general, except the $\sigma$ of the simulated NINO3.4 index from the FGOALS-s2
model. This shows that the simulated variabilities across the models are, in general, in reasonable
agreement with one another.
Figures 2a shows a 101-year running average of time series of globally-averaged simulated
surface temperature for the JJAS season during LM i.e. from CE 850-1849, (henceforth $T_G$), and Figure
2b, the corresponding time series representing the surface temperature over the Indian sub-continent
(henceforth $T_I$). The 101-year running window has been applied to identify the long term changes. We
note that the simulated signals in all the models evolve coherently in time, but with significant spread
across the models.
To tease out the signal more clearly, we calculated the 101-year running mean temporal
*anomalies of the $T_G$*, presented in Figure 2c and $T_I$ in Figure 2d. We see a relatively more coherent inter-
model evolution in the anomalies as compared to the original data (Figure 2a). This indicates a bias in
the mean climatology of one 'outlier' models. Indeed, it is a standard practice in seasonal prediction to
analyse the anomalies of temperature and rainfall, etc. rather than the original data so that the biases in
the climatology do not mask the coherent signals across the models (e.g. Min et al., 2009). Further,
while there are fluctuations in temperature during LM, we see models showing a warming signal during
the MWP (CE 1000-1199) and cooling during the LIA (CE 1550-1749), in a general agreement with the
observations (e.g. Box TS. 5 Figure 1b, Stocker et al, 2013 IPCC 2013; henceforth referred to as TS-
IPCC13). Interestingly, in addition to these two well-known epochs, we see a few more warm and cold
climatic periods, but with a shorter duration.
The spread at the time series of $T_I$ in different models (Figure 2b) is slightly more compared to
that for the $T_G$ (Figure 2a). Figures 2a and 2b indicate that the global mean temperature varies roughly
13°C to 16°C across the models through the LM. The corresponding range for the Indian sub-continent
is 25°C to 29°C. Interestingly, the corresponding temporal anomalies across these models are more
coherent and less-spread out magnitude as compared to the corresponding anomalies of $T_G$ from various
models. The sharp cooling around CE 1250 seen in the global summer temperature across the models is

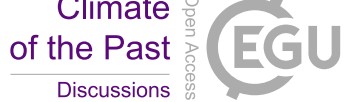



simulated over the Indian region as well (Figures 2c and 2d), and is coincident with a strong volcano
(Gao et al., 2008; Liu et al., 2016; Iles et al., 2014). Such a signal is apparent from a few proxy records
as well (e.g. Fig. 1 of Box TS5, TS-IPCC13). Also evident is that all the modelled temperatures have
apparently entered a cooling phase from this point. We show a proxy record from north India (33ºN,
76ºE; adopted from R. R. Yadav et al., 2009) with the model simulations (Figure A3), which indicates a
qualitative agreement between the simulations and the proxy records.

8       The 101-year running averages of the simulated ISMR anomaly, area-averaged over 65ºE-95ºE

to 10ºN-30ºN, are presented in Figure 3. A linear trend analysis of ISMR during LM (Figure A4) shows
a statistically significant (at 0.1 level) but moderate decreasing trend in four models throughout the LM,
in agreement with findings from several proxy records (e.g. Figure 8 of Ramesh et al., 2010). The MPI
model also shows a weak decreasing trend. In contrast, two models, HadCM3 and IPSL, simulate a
moderate increasing trend. Figure 3 also shows an inter-model spread in the anomalous evolution of the
ISMR through the MWP. As it is, the spread in the simulated IMSR rainfall across the models is known
to be a general limitation of the models (e.g. Jourdain et al., 2013). In comparison, as seen in Figures 2b
and 2d, the simulated temperature response over India during the MWP and LIA is relatively more
coherent across the models, and its evolution qualitatively agrees with the available proxy records
(Yadava et al., 2005; Ramesh et al., 2010; Thamban et al., 2007).
In general, higher (lesser) rainfall as compared to the LM mean is seen during most of the MWP
(LIA) over India in a majority of the models. Table S3 shows that four (five) of the seven models
simulate an anomalously higher (lower) than the mean ISMR during the MWP (LIA). Further, six
models simulate a higher ISMR during MWP as compared to the corresponding historical simulations
(Figure not shown). As corresponding to the average observed mean ISMR for the 1950-2005, the
deficit ISMR during the LIA from three models is about 30% to 40%, a value similar to that suggested
from proxy data analysis (Yadava et al., 2005).
In Table 2a, we show the simulated correlation coefficients between $T_I$ and NINO3.4 index for
the LM period, as well as those in the first, middle and the last 500 year periods of the LM. Similar
correlations between the area-averaged ISMR and NINO3.4 index are presented in the Table 2b. In
general, these simulated NINO3.4-ISMR correlations are negative, while the corresponding NINO3.4-$T_I$
correlations are positive. Importantly, most of the correlations are significant at 0.05 level from a 2-
tailed Student's t-test, suggesting that ENSO has been consistently influencing the Indian climate
throughout the LM. Multi-century model simulation studies by Whittenberg et al. (2009) show multi-





decadal changes in the ENSO statistics. The consistent ENSO-monsoon links over a 1000-year
simulation across many models as shown above reconfirms that the ENSO is indeed an important driver
of the interannual Indian summer monsoon climate variability.
*3.3 MWP and LIA Analysis*

7         From Figure 2, we surmise that Indian sub-continent was also warmer (cooler) during the CE

1000-1199 (CE 1550-1749) as compared to the concurrent global mean temperature, which is
reasonable from the context of its tropical and subtropical location.

11        The simulated interannual standard deviations of JJAS surface temperatures (for both global as

well as the Indian regions), the ISMR and the NINO3.4 index during the MWP and LIA periods are
presented in Table S4. From this table, it is evident that the standard deviations have not apparently
changed much across the LM. The simulated LIA standard deviations, however, are in general
agreement with amplitude of MWP standard deviations.

17        Spatially distributed simultaneous correlation coefficients between the summer NINO3.4 index

with the local summer monsoon rainfall during the MWP and LIA from the individual models are
shown in Figure A5, and those with the corresponding surface temperature are shown in the Figure A6.
Simultaneous area-averaged correlation coefficients between the summer NINO3.4 index with the local
summer monsoon rainfall during the MWP and LIA from the individual models are shown in Figure 4a,
and those with the corresponding surface temperature are shown in the Figure 4b. The signs of the
correlations agree with those of the historical i.e. current period. Further, the magnitudes of all these
correlations are comparable to the corresponding correlations from observations during the historical
period, as well as statistically significant at 0.1 level. Note that the simulated ISMR-NINO3.4 index
correlations for both MWP and LIA periods, except those for the FGOALS-s2 model, are statistically
significant at 0.05 level from a 2-tailed Student's t-test. Notably, for the Indian region, the magnitudes of
the correlations with the ENSO index are stronger in the case of the surface temperature as compared to
the rainfall (Figure 4). Interestingly, in five models out of the seven, the magnitudes of the correlation
coefficients of the NINO3.4 index with the ISMR, and those of the NINO3.4 index correlations with the
JJAS surface temperature over India are weaker in the LIA relative to the MWP. We also carried out a
bootstrapping significance test (1000 simulations) for the ISMR-NINO3.4 correlation; we find that the
results for all seven models are significant at 0.05 level. Further, four out of seven models show weaker
magnitude in correlations during the LIA relative to the MWP (Figure A7a). These correlations indicate



a strong multi-decadal-through-centennial modulation of the association between the ENSO and ISM
during LM. The difference of ISMR-NINO3.4 correlations between the MWP and LIA in four models
(CCSM4, HADCM3, IPSL and MPI) is statistically significant at 0.1 level, as seen from the
bootstrapping tests (Figure A7b)

6       To explore this aspect further, we present the simulated frequencies of El Niños and La Niñas

during the MWP and LIA by the individual models in Table 4. For this calculation, we catalogue a
simulated ENSO event as strong when the amplitude of the NINO3.4 index exceeds $1\sigma$ (Table 3).
Interestingly, a majority of the PMIP3 models in this study indicates more strong El Niños (La Niñas) as
compared to the La Niñas (El Niños) during the MWP (LIA). In addition, even the total (including weak
and strong events) El Niños (La Niñas) are more in MWP (LIA). We also see from Table 4 that all
models except the BCC model consistently simulate more El Niños as compared to La Niñas (including
the strong events) during the MWP compared to the LIA; this result is statistically significant at 0.05
level from a 2-tailed Student's t-test carried out for difference of means. Further, there is relatively more
discrepancy in the difference in the simulated El Niño and La Niña frequencies, i.e. the skewness of
ENSO, across the models in the LIA simulations as compared to those for the MWP. Particularly, the
BCC model simulates a relatively more number of El Niños during the LIA.
In the recent period, El Niños (La Niñas) have been suggested to cause an anomalous increase
(decrease) in global temperature (e.g. Trenberth et al, 2002). Interestingly, as mentioned above, a
majority of the PMIP3 models in this study indicates more El Niños as compared to the La Niñas during
the MWP. A LM simulation study using the CCSM4 model (Landrum et al., 2013) does not simulate La
Niña–like cooling in the eastern Pacific Ocean during the MWP relative to the LIA. In the recent period,
El Niños (La Niñas) have been suggested to cause an anomalous increase (decrease) in global
temperature (e.g. Trenberth et al, 2002). Importantly, a study using a Cane-Zebiak type of coupled
model (Mann et al., 2005) suggests more La Niña-like conditions during the MWP.
In this context, it is pertinent to note that several proxy based studies (Cobb et al. 2003; Graham
et al. 2007; Mann et al. 2009) suggest either a weak ENSO variance or more La Niñas during the MWP.
A study by Henke et al., (2017) based on precipitation proxy data compilation shows a propensity of
more El Niño-like LIA compared to the MWP; however as per Henke et al. (2017), the difference is not
statistically significant and, is not apparent in a proxy-derived temperature compilation. On the other
hand, a study by Conroy et al., (2008), finds that their diatom record is not consistent on SST
interpretation with that of a coral record (Cobb et al., 2003). Specifically, while the diatom record



suggests warmer SST in the eastern equatorial Pacific during some part of the medieval period, the coral
derived SST indicates a cooling trend in the same location. Conroy et al. (2008) suggest a more
heterogeneous SST in the region. Notably, Henke et al. (2017) claim that their result is insensitive to the
choice of definition for the MWP and LIA. Therefore, a higher number of the PMIP3-simulated El
Niños as compared to La Niñas in almost all the models during the MWP is supported to a good extent
by Conroy et al. (2008)'s observations, and reasonably well with the proxy-temperature analysis of
Henke et al. (2017).
Given this agreement across the models, which have a more detailed oceanic component as
compared to simpler models such as that used in Mann et al. (2005), the relevance of any positive
skewness in ENSOs for global temperature during the MWP needs to be verified by making some
AGCM sensitivity experiments, which we plan to do in near future."
Despite the statistically significant correlations between the simulated ISMR-NINO3.4 index, it
will be interesting to explore any non-linearity in the association. When averaged over the seven
models, the percentage of strong El Niño events with concurrent negative ISMR anomalies (henceforth
referred to as EL⁻) is about 70 and 75 during MWP and LIA, respectively (Table 5 and Figure 5). To be
specific, three models simulate a significantly higher proportion of EL⁻ during LIA (89%, 78% and 81%
of strong El Niños in LIA) as compared to those in MWP (69%, 51% and 67% of El Niños in MWP).
Two other models simulate an almost equal number (up to a difference of 1%) of EL⁻. Thus, we can say
that the simulated El Niños during LIA tend to be more 'efficient' as compared to those in MWP in
causing negative ISMR anomalies
On the other hand, it is evident from the Table 5, the model-averaged percentage of strong La
Niñas with positive ISMR anomalies (referred to as LN⁺) shows a higher percentage during MWP
(68%) than during LIA (56%). Five models simulate significantly higher numbers of LN⁺, among all La
Niñas during MWP (75%, 70%, 97%, 57% and 50%) as compared to those in LIA (68%, 55%, 92%,
33% and 42%). One model simulates an almost equal number of LN⁺. Therefore, we infer that the
simulated La Niñas are apparently more 'efficient' during MWP compared to those in LIA causing
positive ISMR anomalies.
We have repeated the analysis for all simulated ENSO events with a magnitude of 0.5 σ, or
above (potentially neither statistically strong nor weak enough to be called as ENSO-neutral) . The
results (not shown) are   qualitatively similar those discussed above.



The above results indicate the propensity of the simulated El Niños (La Niñas) during the LIA
(MWP) to be relatively more 'efficient' in delivering the canonical impact on the summer monsoon
rainfall in India, notwithstanding the statistically significant NINO3.4-ISMR correlations (Figure A5;
Figure 4a). This suggests a possibility of background changes modulating the interannual Indian
summer monsoon rainfall-ENSO association.
*3.4 Possible Dynamics involved – a preliminary analysis*
The large scale Walker circulation is illustrated by the distribution of the anomalous JJAS
velocity potential at the 850 hPa from the NCEP observational analysis for the period CE 1948-1970
(Figure A8) obained from removing the long term (CE 1948-2005) climatology of velocity potential
from CE 1948-1970 velocity potential. The pattern is indicative of a strong convergence over the
western  through the central tropical Pacific region, flanked by a divergence centre to the east, and a
relatively weaker zone of convergence in the Indian Ocean region. After the 1970s, there is a shift in the
Walker circulation (e.g. Vecchi et al., 2006; DeNizio et al., 2013), as seen from Figure A8. The
historical simulations by GISS, IPSL, MPI, CCSM4 and FGOALS-S2 qualitatively simulate the NCEP-
NCAR reanalysis convergence-divergence pattern in the tropical Pacific for the 1948-1970 period
(Figure A8). The low level divergence-convergence pattern in the tropical Indo-pacific simulated by the
HadCM3 and the BCC models (Figure A8) is more reminiscent of that seen from the NCEP-NCAR
reanalysis for the 1971-2005.
Carrying out a detailed analysis of the background dynamics is beyond the scope of the current
study. However, we present results from a preliminary analysis from various models in Figure 6 to
delineate, if possible, the dynamics behind the relatively higher (lesser) rainfall during the MWP (LIA)
over India. Prior to that, we shall briefly explore that the models qualitatively reproducing the zonal
convergence-divergence zones in the tropical Pacific, associated with the Walker circulation, which is
critical for ENSO impacts on climate elsewhere beyond eastern tropical pacific.
Note that, as far as the Figure 6 is concerned, the term 'anomalies' for any parameter during the
MWP (LIA) refers to the excess/deficit of the said parameter during the MWP (or LIA) as compared to
that for the LM (i.e. $P_{MWP}-P_{LM}$, for example, P being any parameter averaged over the corresponding
period). From the distribution of anomalous boreal summer velocity potential at 850 hPa simulated by
CCSM4 shown as example, (Figure 6a and Figure 7) we see a zone of  convergence in the central



tropical Pacific, flanked by two zones of divergence in the equatorial Pacific during the MWP,
suggesting a westward shift in the Walker circulation. We also see a similar shift relative to the
simulations from the historical period (Figure 7). This may suggest a background change  during the
MWP as compared to the LM, and other sub-periods such as the LIA. The anomalous divergence centre
in the west also extends into the equatorial eastern Indian Ocean, which results in an anomalous
convergence zone over India (Figure 6d), and therefore excess rainfall during the MWP (Figure 6c) as
compared to the LM. The corresponding results from the other models are qualitatively similar to those
shown in Figure 6, and available in supplementary material (Figures A9 to A14). The convergence
patterns of the MWP and LIA relative to the historical period (Figure 7) are qualitatively similar to the
anomalous patterns relative to the LM (Figures 7, respectively). Four (five) models simulate anomalous
convergence over India during MWP (LIA) relative to Historical period. The relative patterns over the
tropical pacific are also more or less similar to those from the historical period simulations. In some
models, the extents of the relative convergence/divergence centres are different   from those shown
Figures 6a, 6b, A9 to A14).
We must mention that the composited spatial distribution of rainfall anomalies over the Indian
domain shown in Figure 6 is not statistically significant at 0.1 level from a 2-tailed Student's t-test.
While four other models in addition to the CCSM4, namely GISS, IPSL, HADCM3, and FGOALS-S2,
also show an anomalous excess in rainfall during MWP,  the locations of rainfall surplus over India in
these individual simulations, however,  are not co-located (Figure A15). Having said this,  as a majority
of the models indicates a similar sign of aggregated anomalies in major portions of the region, the
results may qualitatively be considered as conforming across these models. We also see a modest
warming across the region in all simulations of the MWP, in agreement with Figure 2d. The distribution
of temperature anomalies, and their phase, also differs across the models (Figure A15). On the other
hand, during the LIA, the anomalous convergence/divergence (Figure 6b) distribution suggests stronger
convergence in the eastern tropical Pacific compared to the historical period. Interestingly, we also see
an anomalous convergence in the equatorial Indian Ocean, which apparently results in a divergence over
India, and relatively lesser rainfall.
Another factor that is important for the magnitude and variability of the ISMR is the thermal
contrast between the Indian sub-continent and the Indian Ocean during the summer. Recently, a
weakening of land sea thermal gradient had been attributed to a long term weakening trend in the ISMR
(e.g. Sinha et al., 2015; Roxy et al., 2015). We have carried out an analysis of the simulated LSTG
during pre-monsoon i.e. April-May), which is an important factor for the onset and strength of the ISM



(e.g. Pant and Kumar 1997). This is also evidenced by the positive correlations between the LSTG at
850 hPa, derived from the ERA-20CM skin temperature (Hersbach et al. 2015) datasets, with the ISMR
for the period 1901-2005, statistically significant at 0.2 level from a 2-tailed Students t-test (Figure 8a).
To account for the better reanalysis quality, we repeat the analysis for the 1950-1981 period, and these
correlations are significant at confidence 0.1 level (Figure 8a).
Importantly, the signs of the correlations change once the monsoon onset takes place, as
evidenced by the negative correlations between the LSTG-ISMR. During the JJAS season, the
corresponding gradient in the upper atmosphere is supposed to be important (e.g. Roxy et al., 2015;
Goswami et al., 2006).
Coming to the simulations,  we find that  the simulated magnitudes of the pre-monsoon 850 hPa
LSTG vary between 6°-7°C (details not shown), and thus appears to be realistic. Importantly, the
simulated mean pre-monsoon 850 hPa LSTG values for the MWP are higher than those for the LIA in at
least five models.  While such a change is in tune with the relatively higher ISMR during the MWP as
compared to the LIA, the magnitude of the simulated LSTG differences between the MWP and LIA are
very weak[1], hovering around 0.1°C, except a single model showing a corresponding value of 0.2°C
(Figure 8b).  The magnitude of the corresponding difference in the 850 hPa LSTG during JJAS is also
weak (figures not shown). The results from a parallel analysis of the pre-monsoon and monsoonal 200
hPa LSTG (Figures not shown) also appear to be similar. Based on all the above discussion, we can sum
up that the long term changes in LSTG may not have contributed substantially to the changes in the
simulated Indian monsoonal climate from MWP to the LIA.
Furthermore, it is difficult to say state whether such weak changes in the meridional gradient in
the temperature are related to the decadal background circulation changes in the tropical Indo-pacific, or
independent of them; we cannot also comment whether such changes are either commensurate with any
strong external forcing, such as more volcanic eruptions during the LIA, unless we conduct sensitivity
experiments with AGCMs. Unfortunately, carrying out such experiments is beyond the scope of the
current study.
**4. Conclusions and scope for future studies**

---

[1]      The corresponding standard deviation of the simulated 850 hPa LSTG range from 0.7°-1.2°C,
depending on the area over which the pre-monsoon temperatures were calculated (Figures A15a,A15b).



The global climate has experienced significant centennial climate variations in the last two
millennia, without any apparent commensurate changes in the anthropogrnic climate forcing except for
the end of 20$^{th}$ century (IPCC, 2013). Proxy-data based studies identify two significant periods in the
last millennium (LM): (i) a relatively warm period known in the literature as the 'Medieval Warm
Period' (MWP, CE 950-1350) followed roughly after 150-200 years by (ii) a relatively cooler period
referred to as the Little Ice Age (LIA, CE 1500-1850. Notably, variability of ISM in reference to the
above mentioned climatic events is relatively less studied on centennial to millennial time scales. A few
proxy records also document such periods in the Indian region, though the paucity of data introduces
uncertainty in quantifying the climate state parameters during those events.
To complement the proxy-studies, we carry out an analysis of the PMIP3 data sets. We use
available datasets from seven models. We find that the multi-model mean simulates the temperatures
during the MWP and LIA epochs during CE 1000-1199 and CE 1550-1749 roughly commensurate with
the proxy-observations. Our analysis of the PMIP3 data sets suggests that the Indian region was likely
warmer than the global temperature during the MWP. The models also suggest a cooling signal in India
during the LIA.
A majority of the models qualitatively reproduces a wetter (drier) Indian summer monsoon
season in the MWP (LIA) relative to the mean Indian summer monsoon during the LM. The models
simulate a statistically significant ENSO-Monsoon association during the LM in comparison to the
current day climate. Interestingly, we find a propensity of the simulated strong El Niños (La Niñas)
during LIA (MWP) having a relatively more 'efficient' canonical impact, notwithstanding the
statistically significant NINO3.4-ISMR correlation, suggesting a possibility of slow background changes
resulting in an apparent modulation of the interannual ISMR-ENSO association. Indeed, we find a
multi-centennial modulation of the simulated ENSO-ISMR correlations. At least four models suggest a
decreasing ENSO-ISMR (as well as that with the Indian summer temperatures) correlation in the last
500-years of the LM as compared to the first 500-years of LM. Six out of seven models simulate more
El Niños during MWP as compared to La Niñas. Despite such a relatively high occurrence of El Niños
relative to the LM, a relatively westward shift in the simulated summer Walker circulation in
comparison with the mean LM condition is seen in most of the models. The multi-decadal/centennial
shift is reflected in an apparent anomalous divergence in the equatorial eastern Indian Ocean, which in
turn results in anomalous convergence and excess rainfall in the Indian region. Some model studies (e.g.
Ashok et al. 2004) indicate that a presence of anomalous low level divergence in the eastern equatorial
Indian Ocean is critical in causing an anomalous divergence over the peninsular Indian region and





thereby leading to less than mean rainfall there. It is reasonable that the convergence/divergence patterns
in the eastern equatorial Indian Ocean, which is more of a peripheral region for ENSO impact, may
change depending on the background changes in circulation. We must be mindful, however, that the
relatively higher precipitation over India is simulated only in five models, and the location of this excess
precipitation is not the same across these five models. The simulated spatial distribution of the surface
temperature over India is only modestly higher as compared to the corresponding LM average, owing to
the spread of the signals across the models. A plausible reason, which has not been ascertained in this
study, is that the simulated Indian summer rainfall during the MWP mostly comes from a number of
extreme rainfall events as compared to the LM-average, a situation somewhat analogous to warmer and
wetter scenario due to the increased saturated water vapour associated with increased temperature in the
background of global warming (e.g. Lehmann et al., 2015; Goswami et al., 2006). One also needs to be
sensitive to the plausibility that at least some of the changes in the Indian climate (and such changes in
several other regions) during the LM may also be due to 'direct' impacts of the changes in the radiative
forcing through the LM, rather than just due to the 'internal' variability such as the changing ENSO
characteristics. We plan to conduct a suite of atmospheric GCM experiments in addition to some
specially designed coupled experiments in this connection.

18       Further, seven out of seven models simulate more El Niños as compared to La Niñas in MWP

and six out of seven models simulate more La Niñas in LIA as compared to the El Niños. In these
simulations, we see anomalous convergence in the tropical Indian Ocean during the LIA relative to the
LM period, which results in anomalous divergence over the Indian region associated with less summer
rainfall as compared to the corresponding LM mean value. The results, of course are subject to the
model uncertainties and inter-model spread. Having said this, a qualitative agreement across the models,
and the agreement with the findings from available proxy data, gives us some confidence in the results.
It will be interesting to examine, in more detail, the mechanism/reasons for the simulated distinct
summer Walker circulation signatures in the tropical Indian ocean during the MWP & LIA. We also
carry out an analysis of the changes in the simulated pre-monsoon and monsoonal season temperature
gradient between the area-averaged land temperatures in the Indian region and the ocean to its south.
While the results suggest a weakening of such temperature gradient from the MWP to the LIA in
majority of the models, the changes are very weak in magnitude. Another important, relevant aspect that
we hope to study is to explore whether the models are able to simulate the shrinking of the 'Indo-
Pacific' rain belt during the LIA as documented in Denniston et al. (2016) from proxy-data sets, and if
they do, whether such a shrinking has a role to play in the changed ENSO-Monsoon links, at least in the
model world.



**Data Availability**

Model simulation outputs have been downloaded and available from "http://cera-www.dkrz.de/WDCC/ui/Index.jsp".

**Acknowledgements**

We thank Dr. Johann H. Jungclaus, the Max Planck Institute for Meteorology (MPI-M) Hamburg for sharing the model outputs. Useful comments from editor Prof. Hugues Goosse, Dr. Oliver Bothe, and Dr. Wenmin Man on an earlier version of the manuscript are appreciated. The GrADS, ferret and NCL graphics tools, and the CDO statistical software, have been used in this study.

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



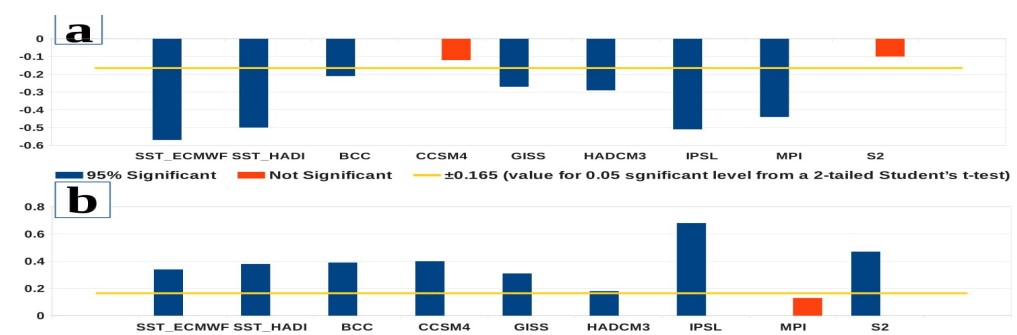

**Figure 1** Correlations, from historical data, between the NINO3.4 and (a) ISMR (b) near-surface air temperature over India (yellow line represents the 0.1 significant level from a 2-tailed Student's t-test), Blue (Red) colour bars show the significant (insignificant) values.

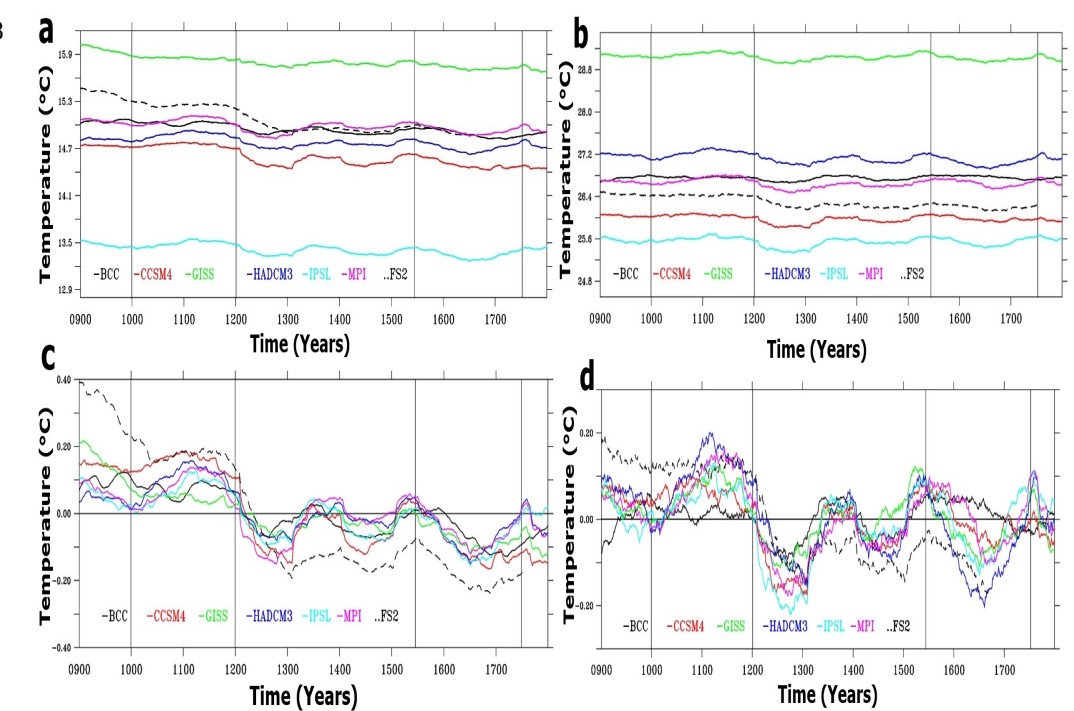

**Figure 2** 101-year running mean of near surface air temperature (°C), obtained by area-averaging (a) globally (b)over the Indian Region, and corresponding anomalies in (c) and (d), respectively.




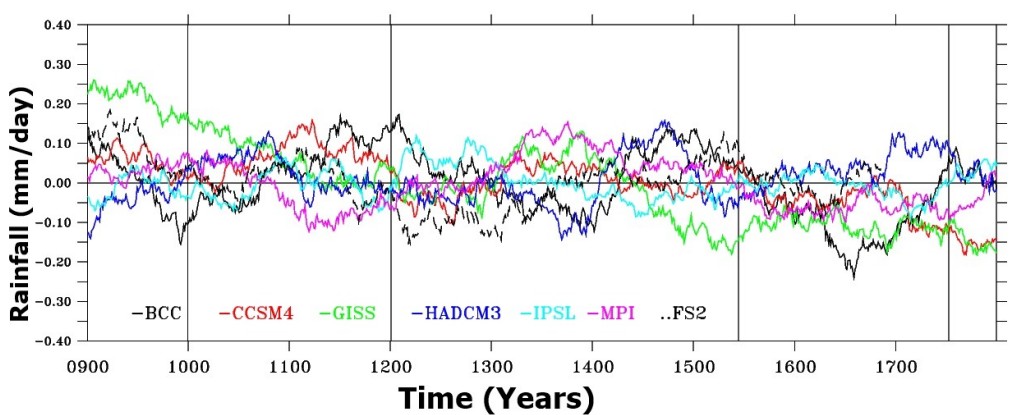

**Figure 3** 101-year Running mean of ISMR anomaly (mm/day).

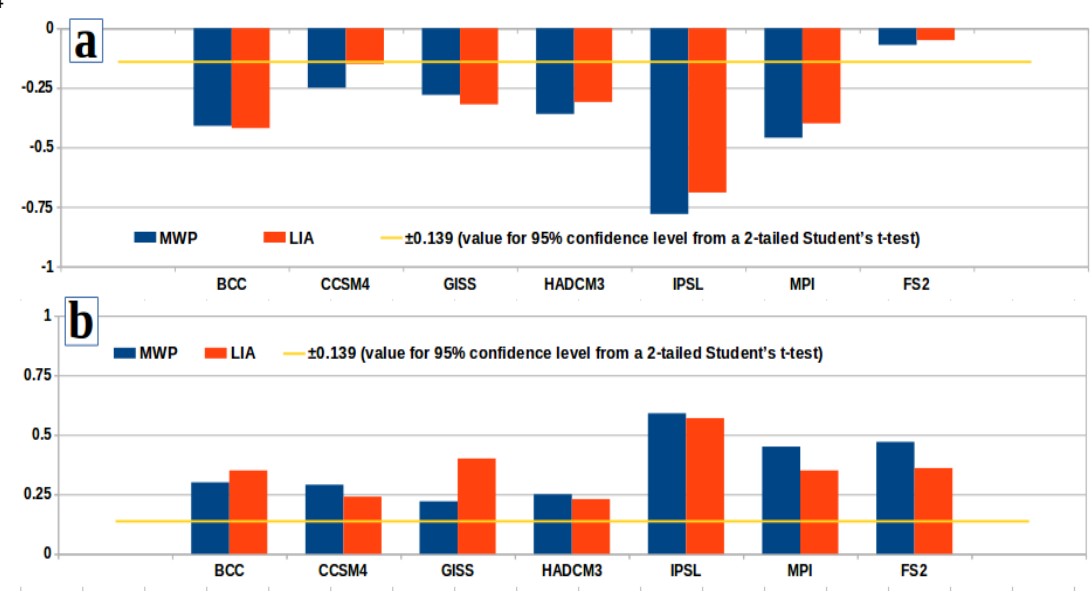

**Figure 4** JJAS Correlations during MWP and LIA between (a) NINO3.4 and ISMR (b) NINO3.4 and Near Air Surface Temperature (yellow line shows the significant value at 0.05 level from a 2-tailed Student's t-test).






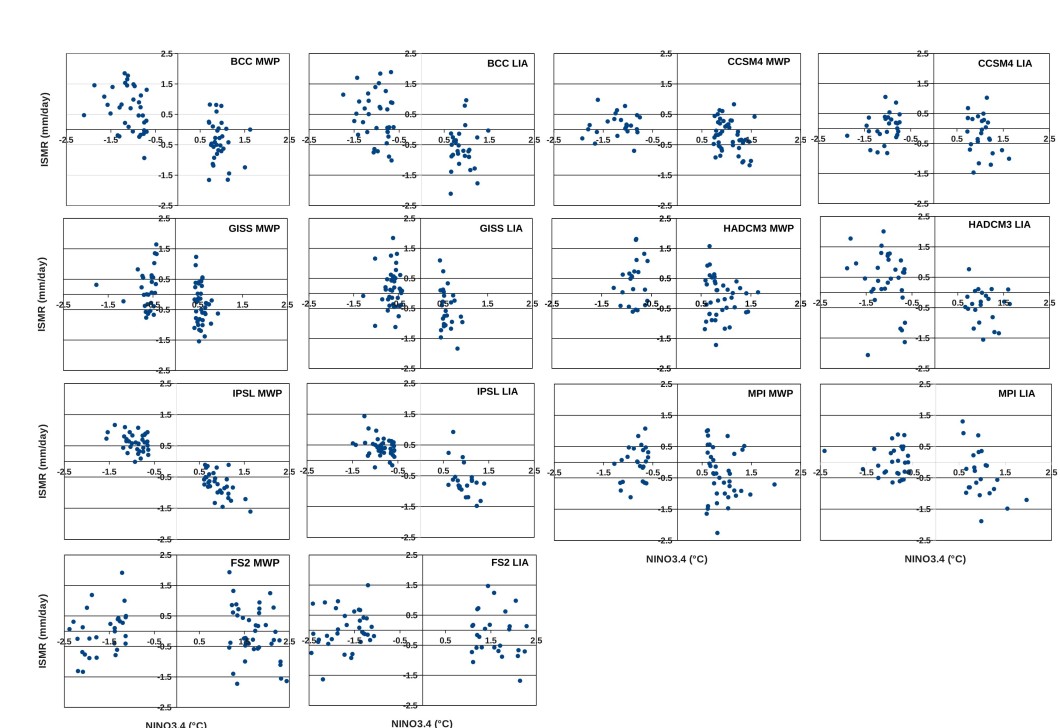

**Figure 5** Scatter diagram showing simulated NINO3.4 (°C) index (X-Axis) and simulated ISMR in
(mm/day Y-Axis) during both MWP and LIA for CMIP5/PMIP3 models. The last descriptor string
in each panel indicates the name of the model and the period (MWP or LIA).





**Figure 6** Distributions of anomalous JJAS rainfall (mm/day; contours) 850 hPa divergent winds (m
s- 1) and velocity potential ( m 2 s- 1; Shaded) from the CCSM4 (a) during MWP-LM, (b) during
LIA-LM, and (c) the respective differences between the MWP & LIA (MWP-LIA). Figures (d), (e)
& (f) are same as Figures 8(a), (b) and (c), respectively, except that they are zoomed into the Indian
& tropical Indian Ocean regions.



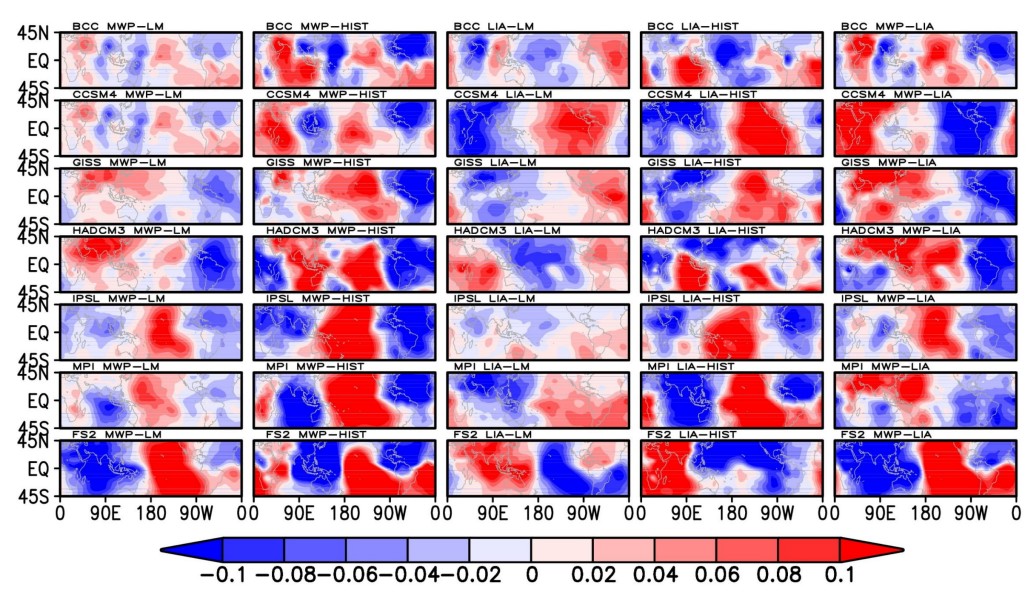

**Figure 7** Distributions of simulated 850 hPa anomalous velocity potential (m²s⁻¹ ) differences
between the MWP (AD 1000-1199)-LM (AD 0850-1849), MWP-Historical Period (1948-2005),
LIA(AD 1550-1749)-LM, LIA-Historical Period and  MWP-LIA over tropical region. The
descriptor string above each panel indicates the name of the model and the period.

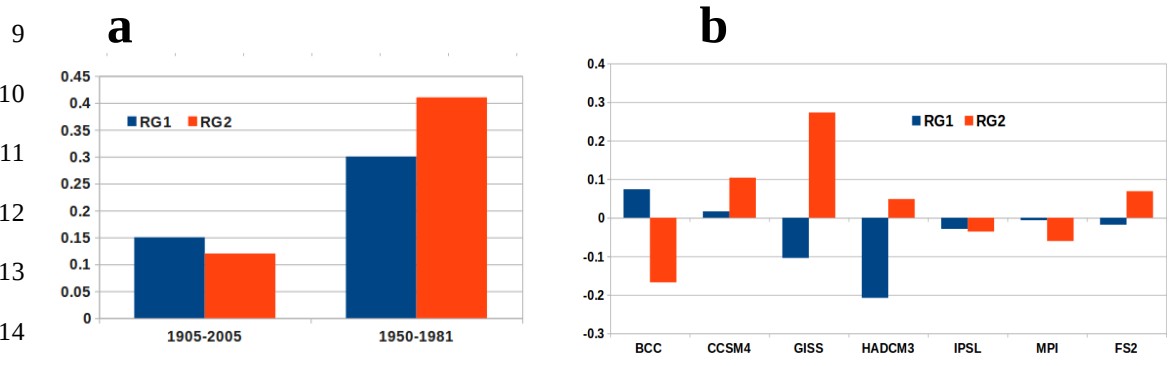

Figure 8 (a) Correlations during 1905-2005 and 1950-1982 between LSTG (April-May) and ISMR
area-averaged over regions RG1 and RG2. (b) Difference of the corresponding simulated LSTG
(April-May) between MWP and LIA (MWP-LIA), area-averaged over the regions RG1 and RG2.
The region RG1 covers most of the Indian land region encompassed by 70°E-90°E, 5°N-35°N (e.g.
Roxy et al., 2015), and (RG2) a land region 65°E-80°E,25°N-35°N.



## 1 Appendix Figures:

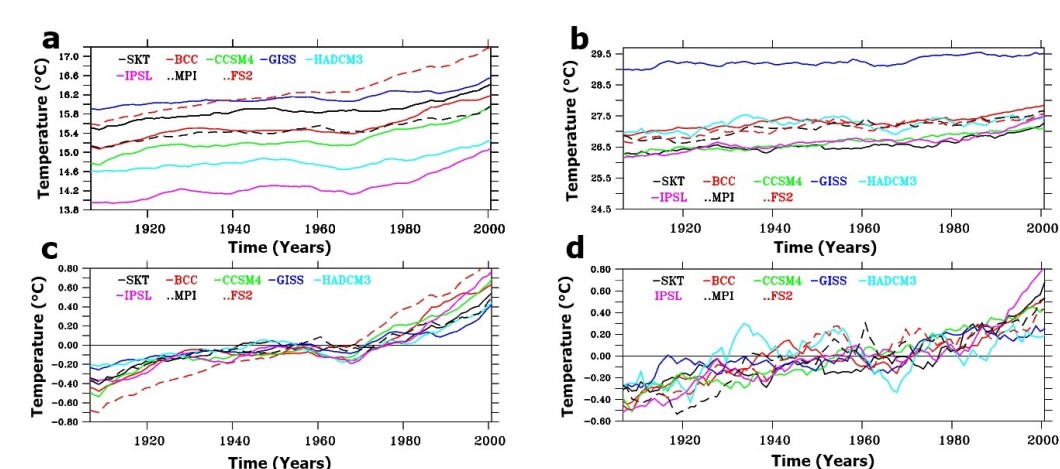

**Figure A1** 11-year running mean surface air temperature (°C), obtained by area-averaging (a)
globally (b) over India; the corresponding temperature anomalies (°C) are shown in (c) and (d),
respectively.

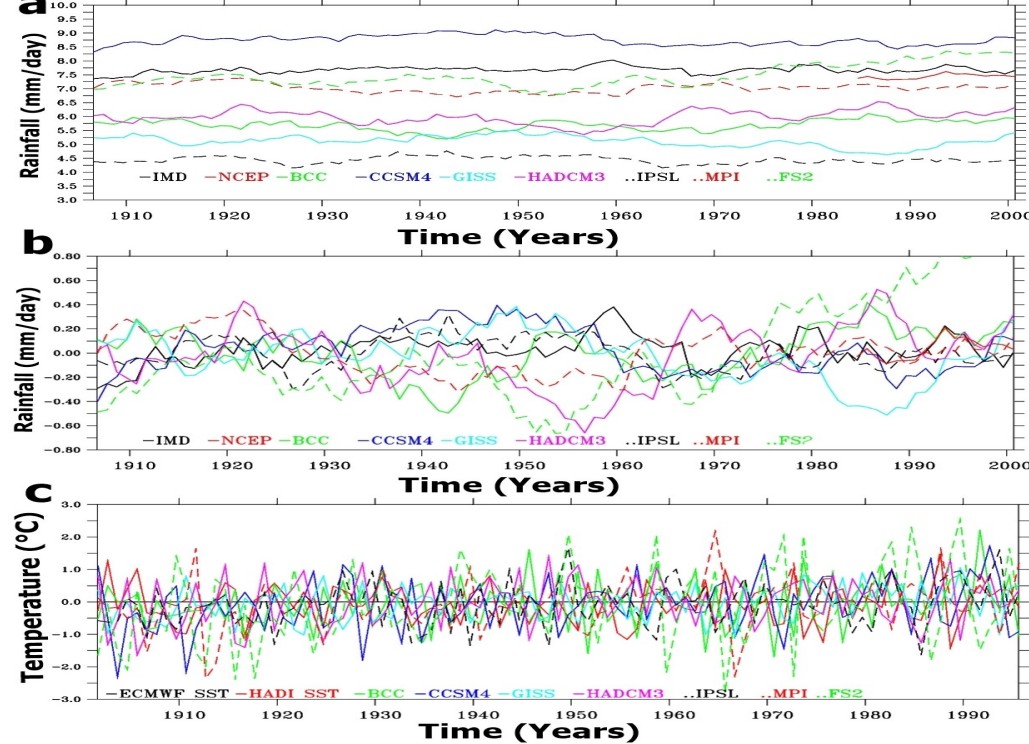

**Figure A2** (a) 11-year running mean of ISMR (mm/day) during the 1901-2005 (b) corresponding
anomaly (mm/day) and (c) JJAS NINO3.4 Index for CE1901-2005.





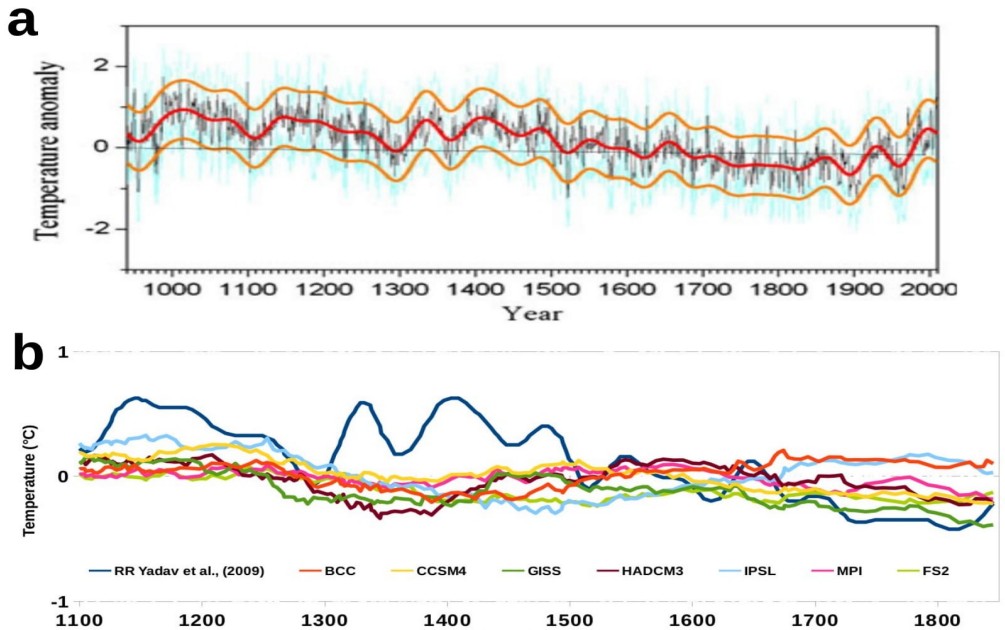

**Figure A3** (a) Mean annual summer (MJJA) temperature reconstruction for the western Himalaya
(33ºN, 76ºE; AD 940–2006). Reconstruction as well as lower and upper one standard errors were
smoothed using 50year low pass filter (R. R. Yadav et al., 2009). (b) Comparision of (a) proxy data
with PMIP3 modelsimulations with 51-year running mean.

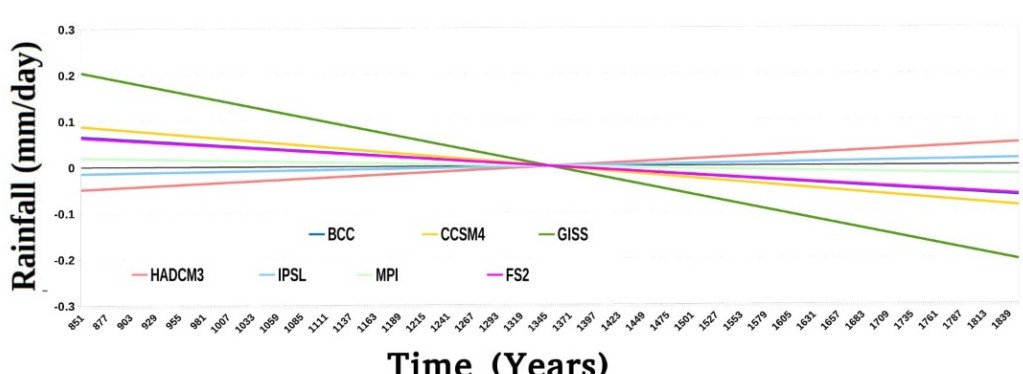

**Figure A4** Linear trend lines of ISMR during LM.





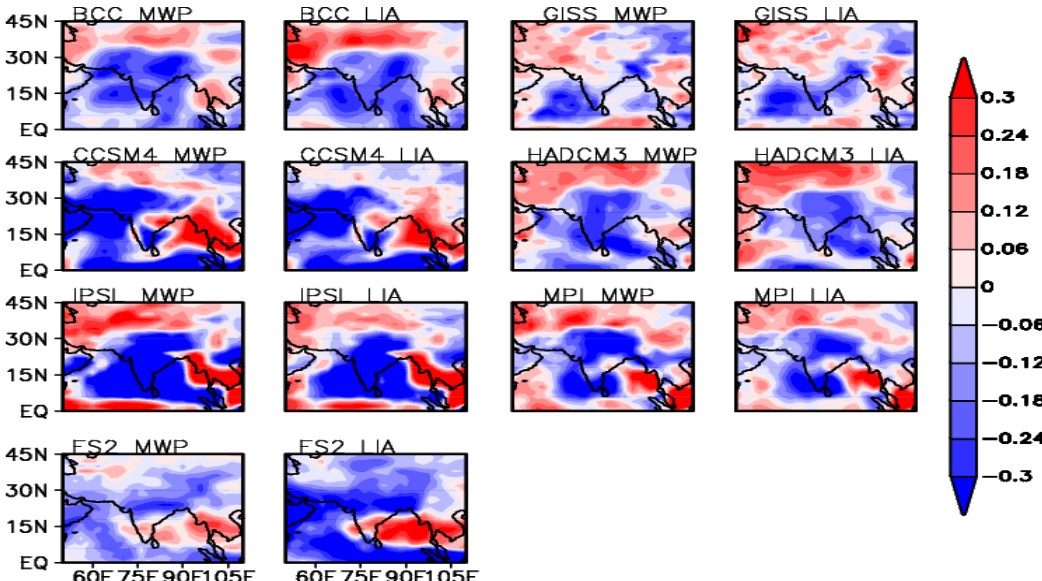

**Figure A5** Spatial distribution of simultaneous correlations for the JJAS season during MWP and
LIA between NINO3.4 and Local Indian Summer Monsoon Rainfall. The  descriptor string above
each panel indicates the name of the model and the period (MWP or LIA).

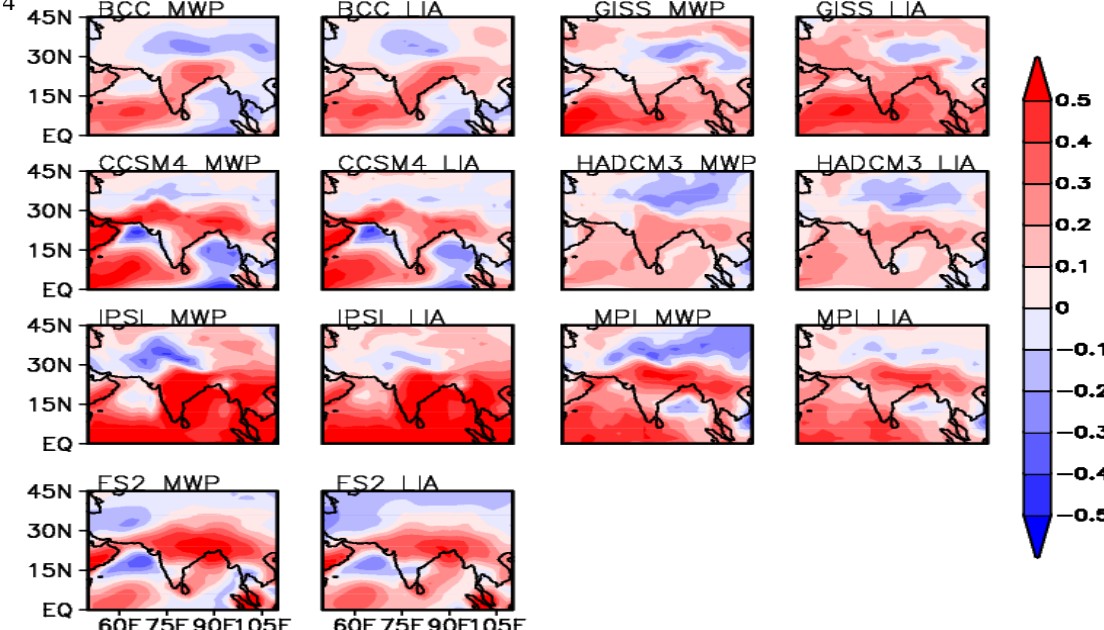

**Figure A6** Spatial plot of simultaneous correlations during MWP and LIA (JJAS) between
NINO3.4 and Local Near Air Surface Temperature. The  descriptor string above each panel
indicates the name of the model and the period (MWP or LIA).




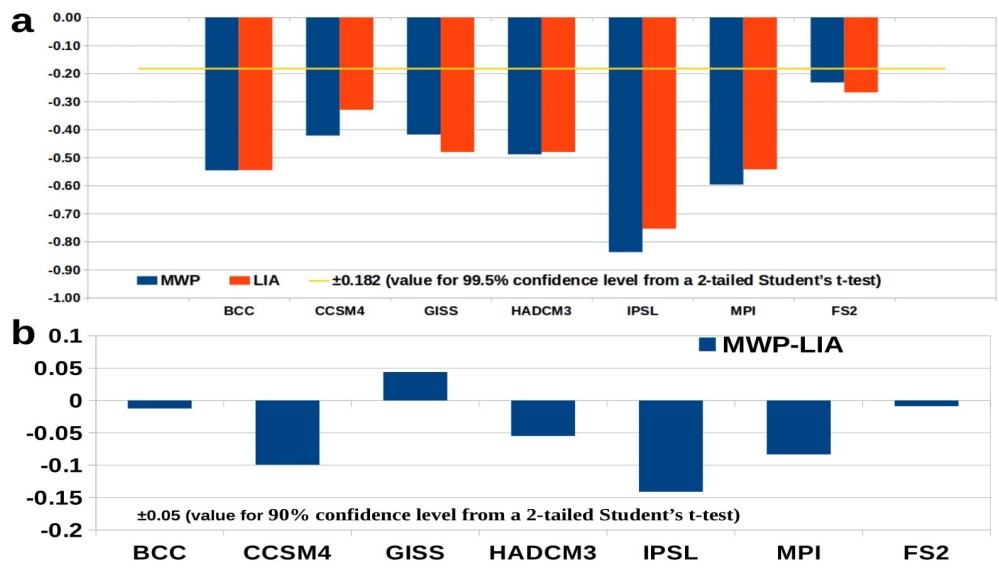

**Figure A7**: (a) Bootstrapping correlations (for 1000 simulations) for 99.5% confidence level during MWP and LIA for individual models. (b) Bootstrapping correlation difference bewtween MWP and LIA (MWP-LIA; for 1000 simulations).

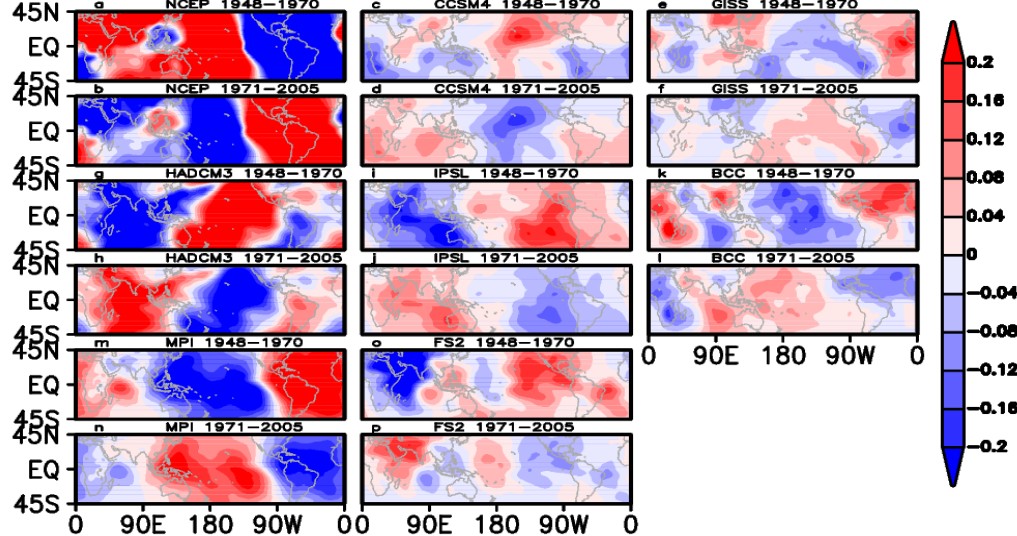

**Figure A8** The panels 'a' and 'b' represents 850 hPa anomalous velocity potential (m2 s-1 ) from the NCEP-NCAR reanalysis for the 1948-1970 and 1971-2008, respectively. The remaining panels are the corresponding results from the historical simulations of various models for these two periods. The descriptor string above each panel indicates the name of the model and the period.





**Figure A9** Distributions of anomalous JJAS rainfall (mm/day; contours) 850 hPa divergent winds (m s- 1) and velocity potential ( m 2 s- 1; Shaded) from the BCC (a) during MWP-LM, (b) during LIA-LM, and (c) the respective differences between the MWP & LIA (MWP-LIA). Figures (d), (e) & (f) are same as Figures 6(a), (b) and (c), respectively, except that they are zoomed into the Indian & tropical Indian Ocean regions.





**Figure A10** Distributions of anomalous JJAS rainfall (mm/day; contours) 850 hPa divergent winds (m s- 1) and velocity potential ( m 2 s- 1; Shaded) from the GISS (a) during MWP-LM, (b) during LIA-LM, and (c) the respective differences between the MWP & LIA (MWP-LIA). Figures (d), (e) & (f) are same as Figures 6(a), (b) and (c), respectively, except that they are zoomed into the Indian & tropical Indian Ocean regions.





**Figure A11** Distributions of anomalous JJAS rainfall (mm/day; contours) 850
hPa divergent winds (m s- 1) and velocity potential ( m 2 s- 1; Shaded) from the
HADCM3 (a) during MWP-LM, (b) during LIA-LM, and (c) the respective
differences between the MWP & LIA (MWP-LIA). Figures (d), (e) & (f) are
same as Figures 6(a), (b) and (c), respectively, except that they are zoomed into
the Indian & tropical Indian Ocean regions.



**Figure A12** Distributions of anomalous JJAS rainfall (mm/day; contours) 850 hPa divergent winds (m s- 1) and velocity potential ( m 2 s- 1; Shaded) from the IPSL (a) during MWP-LM, (b) during LIA-LM, and (c) the respective differences between the MWP & LIA (MWP-LIA). Figures (d), (e) & (f) are same as Figures 6(a), (b) and (c), respectively, except that they are zoomed into the Indian & tropical Indian Ocean regions.





**Figure A13** Distributions of anomalous JJAS rainfall (mm/day; contours) 850 hPa divergent winds (m s- 1) and velocity potential ( m 2 s- 1; Shaded) from the MPI (a) during MWP-LM, (b) during LIA-LM, and (c) the respective differences between the MWP & LIA (MWP-LIA). Figures (d), (e) & (f) are same as Figures 6(a), (b) and (c), respectively, except that they are zoomed into the Indian & tropical Indian Ocean regions.





**Figure A14** Distributions of anomalous JJAS rainfall (mm/day; contours) 850
hPa divergent winds (m s- 1) and velocity potential ( m 2 s- 1; Shaded) from the
FS2 (a) during MWP-LM, (b) during LIA-LM, and (c) the respective
differences between the MWP & LIA (MWP-LIA). Figures (d), (e) & (f) are
same as Figures 6(a), (b) and (c), respectively, except that they are zoomed into
the Indian & tropical Indian Ocean regions.





**Figure A15** Anomalous fields of JJAS surface temperature (°C) and rainfall
(mm/day; shaded) zoomed over Indian region during MWP, during LIA and
MWP-LIA. The descriptor string above each panel indicates the name of the
model and the period.



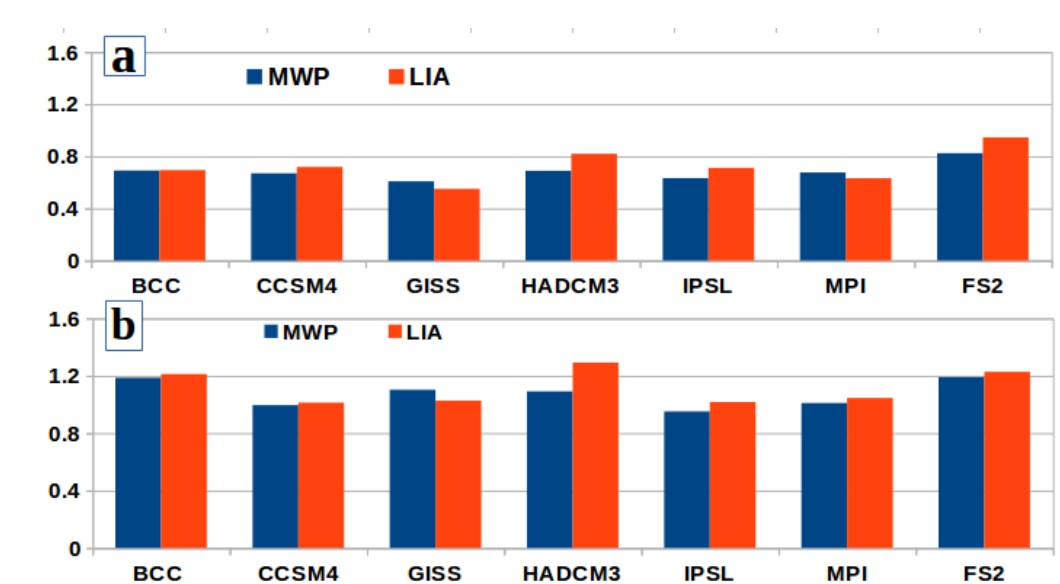

2 **Figure A15** Simulated standard deviation of the LSTG, area-averaged over the
3 regions (a) RG1 (b) RG2 during MWP and LIA





Table 1:- CMIP5/PMIP3 Last Millennium and Historical simulations, their acronyms and temporal
coverage.

| S No | CMIP5/PMIP3 Models | p1000          (Last Millennium) simulation   temporal coverage | Historical          simulation temporal coverage | Acronyms |
|------|--------------------|-----------------------------------------------------------------|--------------------------------------------------|----------|
| 1 | BCC-CSM-1-1(m) | CE 0850-1849 | CE 1850 -2005 | BCC |
| 2 | CCSM4 | CE 0850-1849 | CE 1850 -2005 | CCSM4 |
| 3 | IPSL-CM5A-LR | CE 0850-1849 | CE 1850 -2005 | IPSL |
| 4 | MPI-ESM-P | CE 0850-1849 | CE 1850 -2005 | MPI |
| 5 | GISS-E2-R | CE 0850-1849 | CE 1850 -2005 | GISS |
| 6 | FGOALS-s2 | CE 0850-1849 | CE 1850 -2005 | FS2 |
| 7 | HadCM3 | CE 0850-1849 | CE 1850 -2005 | HADCM3 |

Table 2a:- Correlation between NINO3.4 and ISM surface temperatures during Last Millennium, as
simulated by CMIP5 models (Significant correlation values are shown in **bold** and are significant at
less than 0.05 level from 2-tailed student's t-test).

| S No | Models | CE 0850-1849 | CE 0850-1349 | CE 1100-1599 | CE 1350-1849 |
|------|--------|--------------|--------------|--------------|--------------|
| 1 | BCC | **0.26** | **0.29** | **0.23** | **0.23** |
| 2 | CCSM4 | **0.31** | **0.36** | **0.39** | **0.26** |
| 3 | GISS | **0.38** | **0.31** | **0.38** | **0.43** |
| 4 | HADCM3 | **0.30** | **0.31** | **0.30** | **0.28** |
| 5 | IPSL | **0.59** | **0.61** | **0.58** | **0.58** |
| 6 | MPI | **0.47** | **0.48** | **0.49** | **0.47** |
| 7 | FS2 | **0.41** | **0.44** | **0.37** | **0.35** |




Table 2b:- Correlation between NINO3.4 and ISMR during Last Millennium, as simulated by CMIP5/PMIP3 models (Significant correlation values are shown in **bold** *(italic)* and are significant at less than 0.05 (0.10) level from 2-tailed student's t-test).

| S No | Models | CE 0850-1849 | CE 0850-1349 | CE 1100-1599 | CE 1350-1849 |
|------|--------|--------------|--------------|--------------|--------------|
| 1 | BCC | **-0.32** | **-0.34** | **-0.30** | **-0.29** |
| 2 | CCSM4 | **-0.12** | *-0.08* | **-0.11** | **-0.17** |
| 3 | GISS | **-0.28** | **-0.24** | **-0.33** | **-0.34** |
| 4 | HADCM3 | **-0.39** | **-0.37** | **-0.37** | **-0.40** |
| 5 | IPSL | **-0.70** | **-0.74** | **-0.69** | **-0.66** |
| 6 | MPI | **-0.43** | **-0.43** | **-0.46** | **-0.44** |
| 7 | FS2 | *-0.05* | *-0.07* | *-0.05* | *-0.03* |

Table 3 El Niño and La Niña Classification

| S. No | El Niño Classification | | La Niña Classification | |
|-------|------------------------|---|------------------------|---|
| 1 | 0 < 1σ | Weak El Niño | (-1σ) < 0 | Weak La Niña |
| 3 | > 1σ | Strong El Niño | < (-1σ) | Strong La Niña |

Table 4:- Frequency table of simulated El Niños and La Niñas during MWP (CE 1000-1199) and LIA (CE 1550-1749) of CMIP5/PMIP3 models.

| S N o | Models | MWP (CE 1000-1199) | | | | LIA (CE 1550-1749) | | | |
|-------|--------|---------------------|----------------------|-------------------|---------------------|---------------------|-----------------------|-------------------|---------------------|
| | | Weak El Niños | Strong El Niños | Weak La Niñas | Strong La Niñas | Weak El Niños | Strong El Niños | Weak La Niñas | Strong La Niñas |
| 1 | BCC | 83 | 33 | 48 | 36 | 79 | 29 | 57 | 35 |
| 2 | CCSM4 | 68 | 45 | 63 | 24 | 81 | 27 | 60 | 29 |
| 3 | GISS | 64 | 42 | 64 | 30 | 72 | 28 | 59 | 41 |
| 4 | HADCM3 | 74 | 41 | 62 | 23 | 69 | 23 | 74 | 34 |
| 5 | IPSL | 78 | 32 | 56 | 34 | 73 | 23 | 64 | 40 |
| 6 | MPI | 69 | 40 | 65 | 26 | 55 | 33 | 79 | 39 |
| 7 | FS2 | 75 | 41 | 54 | 30 | 58 | 27 | 79 | 35 |

1                                                    42



1     Table 5 Percentage analysis of 'strong' (a) El Niños with positive and negative ISMR anomalies (EL⁻),

2     and (b) La Niñas with positive an negative ISMR anomalies (LN⁺) during both MWP and LIA from

3     Table-3.

| Models | MWP EL+ | LIA EL- | MWP EL- | LIA EL- | MWP LN+ | LIA LN+ | MWP LN- | LIA LN- |
|---|---|---|---|---|---|---|---|---|
| BCC | 30 | 10 | 70 | 90 | 75 | 68 | 25 | 31 |
| CCSM4 | 33 | 44 | 67 | 55 | 71 | 55 | 29 | 445 |
| GISS | 29 | 21 | 79 | 78 | 56 | 58 | 43 | 41 |
| HADCM3 | 49 | 22 | 51 | 78 | 69 | 76 | 30 | 23 |
| IPSL | 00 | 13 | 100 | 87 | 97 | 91 | 3 | 8 |
| MPI | 32 | 18 | 68 | 82 | 57 | 33 | 42 | 67 |
| FS2 | 44 | 48 | 56 | 52 | 50 | 43 | 50 | 57 |
| AVERAGE | 31 | 25 | 70 | 75 | 68 | 61 | 31 | 39 |

Positive (+) = Positive anomalies of ISMR

Negative (-) = Negative anomalies of ISMR

EL+(-)= Positive (Negative) ISMR Anomalies associated with El Niños

LN+(-)= Positive (Negative) ISMR Anomalies associated with La Niñas



**Appendix Tables:**
Table A1:- Interannual standard deviation of observational and historical simulations of area-averaged
near air-surface temperature over global (TASG) and Indian region (TASI) (◦C), NINO3.4 index (◦C),
and area-averaged Indian summer monsoon rainfall (ISMR), defined as the Observed/Reanalysis data
and Historical simulations.

| S No | Models/Observations | Variables | | | |
|---|---|---|---|---|---|
| | | TASG (ºC) | TASI (ºC) | NINO3.4 Index (ºC) | ISMR (mm/day) |
| 1 | SST_HADI | NA | NA | 0.60 | 0.69 (RF_IMD) |
| 2 | SST_ECMWF | NA | NA | 0.70 | 0.53 (PRECIP_NOAA) |
| 3 | BCC | 0.33 | 0.36 | 0.76 | 0.77 |
| 4 | CCSM4 | 0.35 | 0.43 | 0.80 | 0.60 |
| 5 | GISS | 0.20 | 0.34 | 0.52 | 0.68 |
| 6 | HADCM3 | 0.22 | 0.52 | 0.71 | 0.82 |
| 7 | IPSL | 0.36 | 0.47 | 0.71 | 0.59 |
| 8 | MPI | 0.26 | 0.48 | 0.74 | 0.57 |
| 9 | FS2 | 0.49 | 0.44 | 1.19 | 0.83 |

Table A2:- Boreal summer interannual standard deviation of near air area-averaged surface
temperature over the globe (TASG) and that over India (TASI), and that of ISMR, as simulated by
CMIP5/PMIP3 Last Millennium models (here A: CE 0850-1849; B: CE 0850-1349; C: CE 1100-1599
and D: CE 1350-1849)

| S No | Models | Variables | | | | | | | | | | | | | | | |
|---|---|---|---|---|---|---|---|---|---|---|---|---|---|---|---|---|---|
| | | TASG (ºC) | | | | TASI (ºC) | | | | ISMR (mm/day) | | | | NINO3.4 Index (ºC) | | | |
| | | A | B | C | D | A | B | C | D | A | B | C | D | A | B | C | D |
| 1 | BCC | 0.13 | 0.13 | 0.13 | 0.11 | 0.29 | 0.30 | 0.29 | 0.28 | 0.76 | 0.74 | 0.75 | 0.77 | 0.65 | 0.65 | 0.64 | 0.65 |
| 2 | CCSM4 | 0.25 | 0.25 | 0.27 | 0.21 | 0.38 | 0.40 | 0.40 | 0.37 | 0.62 | 0.59 | 0.60 | 0.64 | 0.73 | 0.75 | 0.74 | 0.72 |
| 3 | GISS | 0.19 | 0.18 | 0.16 | 0.18 | 0.35 | 0.33 | 0.35 | 0.36 | 0.70 | 0.70 | 0.71 | 0.69 | 0.45 | 0.43 | 0.45 | 0.46 |
| 4 | HADCM3 | 0.20 | 0.19 | 0.19 | 0.20 | 0.46 | 0.33 | 0.46 | 0.49 | 0.76 | 0.73 | 0.74 | 0.78 | 0.63 | 0.60 | 0.62 | 0.60 |
| 5 | IPSL | 0.19 | 0.20 | 0.20 | 0.18 | 0.39 | 0.41 | 0.41 | 0.37 | 0.54 | 0.55 | 0.56 | 0.53 | 0.60 | 0.60 | 0.62 | 0.60 |
| 6 | MPI | 0.20 | 0.20 | 0.21 | 0.20 | 0.42 | 0.41 | 0.44 | 0.42 | 0.60 | 0.61 | 0.60 | 0.69 | 0.59 | 0.60 | 0.63 | 0.58 |
| 7 | FS2 | 0.26 | 0.25 | 0.21 | 0.18 | 0.39 | 0.40 | 0.40 | 0.36 | 0.75 | 0.76 | 0.76 | 0.74 | 1.14 | 1.15 | 1.15 | 1.11 |



Table A3: Percentage (%) of Increase/Decrease in ISMR relative to Observation (CE 1950-2005) and respective LM-
mean, Historical-mean (CE 1950-2005).

| S. No | Models | Percentage (%) of Increase/Decrease in ISMR relative to respective model LM-mean (CE 0850-1849) | | | Percentage (%) of Increase/Decrease in ISMR relative to respective Historical simulation (CE 1950-2005) | | | Percentage (%) of Increase/Decrease in ISMR relative to observation (CE 1950-2005) | | |
|---|---|---|---|---|---|---|---|---|---|---|
| | | MWP | LIA | MWP-LIA | MWP | LIA | MWP-LIA | MWP | LIA | MWP-LIA |
| 1 | BCC | 0.90 | -1.97 | 2.87 | -1.88 | -4.65 | 2.77 | -26.24 | -28.32 | 2.08 |
| 2 | CCSM4 | 0.67 | -0.23 | 1.00 | 1.96 | 1.04 | 0.92 | 15.67 | 14.62 | 1.05 |
| 3 | GISS | 0.94 | -2.07 | 3.01 | 5.74 | 2.59 | 3.15 | -30.02 | -32.15 | 2.13 |
| 4 | HADCM3 | 0.31 | 1.10 | -0.79 | 6.32 | 7.15 | -0.83 | -16.32 | -15.67 | -0.65 |
| 5 | IPSL | -0.22 | -0.22 | 0.00 | 1.65 | 1.65 | 0.00 | -41.25 | -41.25 | 0.00 |
| 6 | MPI | -0.13 | -0.91 | 0.78 | 9.40 | 8.55 | 0.85 | 0.26 | -0.52 | 0.78 |
| 7 | FS2 | -0.14 | 2.50 | -2.36 | -5.22 | -2.74 | -2.48 | -5.22 | -2.75 | -2.47 |

Table A4:- Boreal summer simulated interannual standard deviation for area-averaged near air surface
temperature over Global region (TASG) and Indian region (TASI), area-averaged Indian summer
monsoon rainfall (ISMR) and NINO3.4 Index during MWP (CE 1000-11199) and LIA (CE 1550-
1749) of CMIP5/PMIP3 models.

| S No | Models | Variables | | | | | | | |
|---|---|---|---|---|---|---|---|---|---|
| | | TASG (ºC) | | TASI (ºC) | | ISMR (mm/day) | | NINO3.4 Index (ºC) | |
| | | MWP | LIA | MWP | LIA | MWP | LIA | MWP | LIA |
| 1 | BCC | 0.11 | 0.13 | 0.28 | 0.29 | 0.76 | 0.82 | 0.70 | 0.64 |
| 2 | CCSM4 | 0.13 | 0.17 | 0.33 | 0.32 | 0.53 | 0.60 | 0.73 | 0.72 |
| 3 | GISS | 0.09 | 0.17 | 0.29 | 0.34 | 0.69 | 0.67 | 0.42 | 0.42 |
| 4 | HADCM3 | 0.15 | 0.18 | 0.39 | 0.44 | 0.73 | 0.76 | 0.58 | 0.65 |
| 5 | IPSL | 0.17 | 0.17 | 0.37 | 0.36 | 0.56 | 0.51 | 0.61 | 0.58 |
| 6 | MPI | 0.13 | 0.17 | 0.37 | 0.40 | 0.67 | 0.60 | 0.58 | 0.58 |
| 7 | FS2 | 0.13 | 0.15 | 0.37 | 0.36 | 0.75 | 0.71 | 1.14 | 1.07 |

