# Peer review of "The ENSO teleconnections to the Indian summer monsoon climate through the Last Millennium as simulated by the PMIP3. Charan Teja Tejavath[a], Karumuri Ashok[a], Supriyo Chakraborty[b] and Rengaswamy Ramesh[c]"

_Climate of the Past, 2018_

## Referee Comment (RC1) · O. Bothe (Referee) · 1 Mar 2018

Dear authors,

In your manuscript "The ENSO teleconnections to the Indian summer monsoon climate through the Last Millennium as simulated by the PMIP3" you present results on the relation between the El Niño Southern Oscillation (ENSO) and the climate on the Indian subcontinent during the Indian Summer Monsoon (ISM) season in simulations over the last millennium. You find differences in the occurrence of El Niños and La Niñas between warmer and colder periods during the last millennium, i.e., between your definition of the Medieval Warm Period and the Little Ice Age. You find differences

in ISM rainfall (ISMR) between these two periods. You find different efficiencies of El Niño and La Niña in influencing Indian climate between both periods. You analyse the velocity potential between both periods and identify changes in the Walker Circulation.

As for the two previous versions of the manuscript there are various small issues which in parts have already been identified by the editor in his initial evaluation of your manuscript.

I ask you to also address all the editor's comments in an upcoming revision.

Further, I see one larger problem with the manuscript, which I assume to require a major revision before your work may be considered for final publication.

My comments are:

**1 Major:**

You still use all events larger zero in the analyses leading to Table 4 and related results (see Page 11 Line 6ff). This still incorrectly puts neutral ENSO events in one of the categories of El Niño or La Niña. Please redo this with your latter classification using an 0.5 standard deviation criterion. Relatedly it is unclear what you mean by all El Niños or all La Niñas later in the manuscript thus making it impossible to evaluate whether the description of your results is correct.

**2 Minor:**

Your focus on anthropogenic forcings (e.g., Page 2 Line 8) appears as if you are ignoring natural forcings. Overall this is not the case, but by already starting the introduction this way it gives the reader this impression.

P3 L32: As in previous versions, it remains unclear which modelling studies you mean here.

P4 L1: You do not really reconcile simulations and proxies.

P6 L12: As your bootstrapping procedure still remains unclear, I think the reader needs more details.

Please ensure that Figures which are essential for your argument are part of the general manuscript while those only of supporting relevance are in the appendix. For example, is Figure A1 correctly placed in the Appendix or should part of it be in the main manuscript. Similarly, is Figure A2 essential or not? Please check this for all Figures.

P6 L21ff: I don't understand the reference in this sentence. If I understand it correctly, you refer to the Figures of Stocker et al. (2013) and IPCC (2013). Then you should put them as, e.g., Stocker et al. (2013, their Figures ....).

P6 21ff (and elsewhere): I still don't get the point of most of your discussions of standard deviations. This is especially the case here, where I don't see whether this is your result or an IPCC-result or a result of Stocker et al.

P6 L28: One cannot really identify the trends from this Figure.

P7 L1: This is hardly visible in the Figure.

P7 L17: I cannot follow your writing of "seven (five) models". Figure 1 shows 6 significant models - though it's unclear at which level.

The legend for Figure 1 and its caption contradict each other with respect to which significance level is shown.

P7 L30: Maybe it would help to state your criteria earlier.

P8 L5: I think the formulation "of the corresponding statistic" is ambiguous. Maybe try

to clarify.

P8 L10: I'd like to note again as for the previous version: If I see it correctly, you use the GISS data uncorrected (compare https://www.clim-past-discuss.net/8/C393/2012/cpd-8-C393-2012.pdf), I don't think that's too much of a problem, but you may note this somewhere. Sorry, that I didn't note this in my first review.

Paragraph: P8 L17ff:
Calculating anomalies is a standard procedure. You don't tease out the signal with it. You just visualize the evolution more clearly.
The reference to seasonal prediction is unnecessary.
If there are outliers, then there are two outliers viewed globally. However, I don't think you validly can call them outliers especially considering that the models' 20th century evolution are much closer to each other than the past millennium data suggest. I also wouldn't speak of bias here.

P8 L25: You do not show this agreement with (paleo-)observations.

Paragraph P8 L29ff: The discussion of the climatology-temperatures are interesting, but don't serve a purpose. We learn something about the models, but not about your topic. Further, line 32, it is not "interestingly", but obviously that the anomalies align better - you removed the climatological mean from the series.

P9 L1: We know which volcanic eruption it likely was. See https://doi.org/10.1073/pnas.1307520110, https://www.nature.com/articles/srep34868, doi:10.1038/ngeo2875. We also know that other volcanic eruptions resulted in potentially decades long cooling episodes. One example of a potential reference is doi:10.1038/nature14565.

P9 L6: Well, some agreement. Are there further paleo-observations to show the agreement?

Figure A3: What is plotted here? Global or Indian temperature? Is panel (a) just copied

[Figure]

from the original publication or did you produce the Figure yourself. It looks strange. If you just adopt it, the publisher probably has to check the copyright?

P9 L12: Do you mean "MPI shows an insignificant weak decreasing trend"?

P9 L15: I wouldn't speak of a limitation in this context. A spread is expected, the difference in signal between different periods may be seen as a limitation.

P9 L22: I only count 5 simulations.

P9 L24: I don't think the comparison of absolute values between different time-periods between historical observations and the simulations is valid. If you also provided the difference between the historical simulations and the historical observations, one could evaluate it against the general precipitation bias.

P10 L7ff: I wonder under which circumstances the Indian subcontinent could have a colder absolute mean temperature than the globe as a whole. That is, I still don't see that this paragraph tells us anything of relevance.

Paragraph: P10 L11: If the standard deviations didn't change much, why is there a need to discuss them?

P10 L29ff: If I understand you correctly, you're saying that the bootstrapping shows that the difference between correlation coefficients is significant in a number of models. Are you saying that the 0.05 in Figure A7 is the significance level? If so, please highlight it more clearly. This is the interesting result of the bootstrapping. Maybe it's just the caption that is unclear.

My major concern implies that I don't think the sentences P11 L10 to L16 are valid as long as they relate to all events.

P11 L14ff: The sentence should qualify that this holds for some simulations. Anyway, I am not sure I can follow your argument or even what you are referring to. Let me summarise what I see from your table - if I counted correctly: In the MWP there are in

all models more El Niño (EN) than La Niña (LN). In the LIA five models have more or equal number of LN than EN. Two have more EN than LN. The absolute difference in numbers between both types of events is larger for six models in the MWP and for one model in the LIA.

P11 L16ff: Didn't you state the BCC result in line 12 already?

P11 L22: I am not sure whether the comparison between long term means and inter-annual variability is helpful.

P14 L16ff: Different resolutions and structures of the models imply that patterns of change are not exactly the same - especially for precipitation.

P14 L26: Your comparison is to the Last Millennium, isn't it? Thus, you cannot diag-nose change relative to the historical period from this Figure, can you?

L14 L26ff: This equatorial Indian Ocean change is hard to identify. It may be helpful to elaborate what you are referring to.

P15 L3: A 0.2 level is rather uncommon.

P15 L8: Which negative correlations are you talking off? You don't show these, do you?

P15 L8: You mention the JJAS season, but do you show anything about the LSTG-ISMR-relationship then?

P16 L1: Well, there are more climate forcings than just the anthropogenic.

P16 L14ff: This sentence is still not relevant.

P16 L20: What do you mean by: "statistically significant . . . in comparison to the current day climate"?

P16 L24ff: You find this modulation in some simulations.

P16 L27: Does this refer to all events? If so, my major concern applies.

P16 L28: Why is it "despite" the occurrence?

P16 L28: Is this relatively high compared to the Last Millennium? Did you present this change relative to the full period?

P17 L5: What do mean by "the spatial distribution . . . is . . . higher"?

P17 L18: Does this relate to all events including neutral? See my major concern.

**3 Technicalities:**

Please improve the quality of the Figures. For example, produce them in the correct aspect ratio and do not rescale them in a graphics software.

Please be consistent in using Figure or Table AN or SN, i.e., please check that you always use the abbreviation A for Appendix.

There are a number of typos etc. but I leave those for the copy-editor. There are also further unclear formulations. Some of these appear to have survived from the last two revisions.

P6 L30: Remove "We revise the text accordingly"

P8 L25: I again think your reference is incorrectly formulated here. There are further instances of this.

P11 L23: This sentence is already in the beginning of the paragraph.

P12 L14: Please check that this paragraph, the captions, and the content of Figure 5 and Table 5 are correct.

Paragraph P13 L23ff. Shouldn't this come before the Paragraph P13 L10ff?

Figure A15: The second Figure A15 should be Figure A16, I guess.

P15 Footnote: This should be Figure A16.

P15 L27: I would write "such as volcanic eruptions" instead of "such as more volcanic eruptions during the LIA".

P17 all: You mix the results for Temperature and Precipitation in a way that may confuse the reader. Restructuring this section may help.

P17 L18: seven out of seven should possibly read all?

Best regards

Oliver Bothe

---

## Referee Comment (RC2) · Anonymous Referee #2 · 23 Apr 2018

Summary

The authors use 7 PMIP3 simulations to investigate the ENSO teleconnection to the Indian Summer Monsoon during the Last Millennium. The Author use the present day period to evaluate the model simulations and compare their results to some of the existing proxy reconstructions. The authors claim that during the Medieval Climate Anomaly the frequency of El Nino events is enhanced whereas during the Little Ice Age La Nina events occur more frequently. Then, they discuss some non-linearity which is unfortunately not presented in an understandable way.

Overall judgment

I see that the authors put a lot of effort to analyze PMIP3 simulations. However, the way how the results are presented and even more importantly the questionable content leads to The manuscript lacks a clear structure; the phrasing is inadequate preventing the reader to understand the content. Furthermore, the main result of the study, that during the Medieval Climate Anomaly (MCA) the frequency of El Nino events is enhanced whereas during the Little Ice Age (LIA) La Nina events occur more frequently is questionable as detailed below. So, I recommend to reject the manuscript from publication in Climate of the Past.

General Comments

I. The manuscript needs a serious proof reading by a native speaker.

II. The structure of the manuscript is not clear, e.g., section 3.4 contains again an evaluation part. Presenting 'preliminary results' in a manuscript makes no sense, either the results are solid and necessary or not (then they shall not be presented). The authors made no clear selection of figures. It looks like the 'randomly' selected eight figures (+ 5 tables) in the main part and put the rest of the analysis made in the appendix (which is 15 figures and 4 tables).

III. The manuscript builds on one main finding, namely an increase of El Nino events during MCA and an increase of La Nina events during LIA. The authors ignore the fact that they use the NINO3.4 index which by definition varies a bit with the global mean signal. Thus, if the global mean temperature due to external forcing is increase the Nino3.4 index will certainly be biased positive and lead to or El Ninos (although the cause is a global signal and not a real change in ENSO). The authors already show in their results that ENSO is NOT changing from the MCA to the LIA as the standard deviation during the periods is the same (see page 10).

IV. All figures are of bad quality.

Technical comments

L20-21: Unclear sentence

L22-28: Awkward and unclear statements.

L28: divergence center of what??

L29: convergence of what??

L30: Connection between the two parts separated by a semicolon is not given.

Page 2:

L6: IPCC (2013) is not an adequate reference here, please use more specific references

L7-9: Unclear statement

L11-15: Missing references of definition of time periods of MWP and LIA also for the variation pf the periods you need to give references.

L18: Be more specific about the regions you are referring to.

Paragraph3: There is no logical connection to the paragraph before

L33-34: Akward sentences, please clarify.

I stop here as the entire manuscript is like the first two pages.

---

## Author Comment (AC1) · 11 May 2018

General Comment:

In your manuscript "The ENSO teleconnections to the Indian summer monsoon climate through the Last Millennium as simulated by the PMIP3" you present results on the relation between the El Niño Southern Oscillation (ENSO) and the climate on the Indian subcontinent during the Indian Summer Monsoon (ISM) season in simulations over the last millennium. You find differences in the occurrence of El Niños and La Niñas between warmer and colder periods during the last millennium, i.e., between your definition of the Medieval Warm Period and the Little Ice Age. You find differences

in ISM rainfall (ISMR) between these two periods. You find different efficiencies of El Niño and La Niña in influencing Indian climate between both periods. You analyse the velocity potential between both periods and identify changes in the Walker Circulation. As for the two previous versions of the manuscript there are various small issues which in parts have already been identified by the editor in his initial evaluation of your manuscript.

Response:

Dear Dr. Oliver Bothe, we greatly appreciate your very useful suggestions which we carefully incorporated, and your continuous encouragement. We carefully revised the manuscript in light of these comments, and believe that the current revision meets your expectations. We have also incorporated the suggestions from Prof. Goosse and reviewer 2. We had submitted the responses already, and repeat them below.

Major Comment:

You still use all events larger zero in the analyses leading to Table 4 and related results (see Page 11 Line 6ff). This still incorrectly puts neutral ENSO events in one of the categories of El Niño or La Niña. Please redo this with your latter classification using an 0.5 standard deviation criterion. Relatedly it is unclear what you mean by all El Niños or all La Niñas later in the manuscript thus making it impossible to evaluate whether the description of your results is correct.

Response:

To avoid any further confusion, we remove the discussion and a relevant table of 'all'-ENSO events. We retain only information related to only those with a magnitude of $0.5\sigma$ threshold and above, and focus mainly on 'strong' ($1\sigma$) events. All discussion about 'neutral' events has been removed.

Minor Comments:

Comment B1:

[Figure]

Your focus on anthropogenic forcings (e.g., Page 2 Line 8) appears as if you are ignoring natural forcings. Overall this is not the case, but by already starting the introduction this way it gives the reader this impression.

Response:

We revised the text. Now it reads as "Interestingly, reports based on a large number of publication points out to significant centennial climate variations during the last two millennia (PAGES 2k Consortium, 2013; TS-IPCC13)." We removed the relevant text about anthropogenic forcing.

Comment B2:

P3 L32: As in previous versions, it remains unclear which modelling studies you mean here.

Response:

Thank you. For better clarity, we cite the relevant references at the end of the erstwhile sentence P3L33. Former sentences P3L32-33 now read as "Thus, the variability of Indian summer monsoon during the LM has been relatively less studied, particularly from the modelling perspective. It is also noticeable that all the model studies cited above (Kitoh et al., 2007; Prasad et al., 2014) primarily employed single GCMs."

Comment B3:

P4 L1: You do not really reconcile simulations and proxies.

Response:

Thank you. We remove the statement.

Comment B4:

P6 L12: As your bootstrapping procedure still remains unclear, I think the reader needs more details.

Response:

We have carried out a bootstrapping test with 1000-simulations by randomizing (through NCL) the model-simulate ISMR & NINO3.4 SSTA for MWP, and those for LIA. The bootstrapping correlations are significant at 0.01 level in both MWP and LIA for all models. The difference of the synthetic correlations between the ISMR & NINO3.4 index from the simulations was ordered as per magnitude to find out the significance levels of the correlation difference. We find that the difference of ISMR-NINO3.4 correlations between the MWP and LIA in four models (CCSM4, HADCM3, IPSL and MPI) is statistically significant at 0.1 level.

In this context, we have modified and added the following text on bootstrapping in methodology in section 2. The new text is "We employ a bootstrapping test with 1000-simulations. We use the bootstrapping subroutine "bootstrap_correl", provided by the freely available NCL package from NCAR <https://www.ncl.ucar.edu/Applications/bootstrap.shtml>. This routine takes the two input timeseries (the model-simulated ISMR & NINO3.4 SSTA for MWP, for example, in our case) for which the correlations need to be obtained. Based on these input series, it generates 1000 timeseries pairs randomly, and computes correlations between each pair. After that, the correlations are ordered as per magnitude. Once this is done, the 5th highest correlation, for example, gives us the 0.005 significance level (i.e. 99.5% confidence level) for the correlations. In case of correlation differences between two simulations, such as the MWP & LIA simulations by the same model, the differences of correlations are ordered as per magnitude to identify the significant threshold values.

Comment B5:

Please ensure that Figures which are essential for your argument are part of the general manuscript while those only of supporting relevance are in the appendix. For example, is Figure A1 correctly placed in the Appendix or should part of it be in the main manuscript. Similarly, is Figure A2 essential or not? Please check this for all

Figures.

Response:

Considering the comments from you and another reviewer, we placed the essential pictures in main manuscript and moved rest of them to supplementary part accordingly. Now the manuscript contains 10 Figures in main part and 6 Figures in supplementary part.

Comment B6:

P6 L21ff: I don't understand the reference in this sentence. If I understand it correctly, you refer to the Figures of Stocker et al. (2013) and IPCC (2013). Then you should put them as, e.g., Stocker et al. (2013, their Figures ....).

Response:

Thank you. We have made changes appropriately.

Comment B7:

P6 21ff (and elsewhere): I still don't get the point of most of your discussions of standard deviations. This is especially the case here, where I don't see whether this is your result or an IPCC-result or a result of Stocker et al.

Response:

Thank you. This part of the text has been revised. Now it reads as "Further, we find that the observed as well as the simulated trends are significantly above the corresponding interannual standard deviations (Figure A1).

Comment B8:

P6 L28: One cannot really identify the trends from this Figure

Response:

We have carried out Mann-Kendall test to identify the trends, which identifies the decreasing trend as simulated by three models, namely, CCSM4, MPI and FS2 are statistically significant. For clarity, we present a new figure (Figure R1) with the observed ISMR for the 1980-2005 when it is prominent and corresponding individual simulated ISMR trends for the analogous period. This figure shows that the models qualitatively reproduce the observed trend in the ISMR.

Comment B9:

P7 L1: This is hardly visible in the Figure.

Response:

Thank you. The figure has been revised. Further, in the revised sentence, we cite Table S1 appropriately along with Figure 2Ac for clarity.

Table S1, reproduced below, shows number of Strong (only events whose amplitude is above $1\sigma$) ENSO events after CE 1950 (Attached as Figure R3).

Comment B10:

P7 L17: I cannot follow your writing of "seven (five) models". Figure 1 shows 6 significant models - though it's unclear at which level.

Response:

We modified the sentence. Now it reads as "Corresponding correlations for seven (five) models are statistically significant at 0.1 (0.05) level from a 2-tailed Student's t-test, though they vary over a wide range of values varying from 0.13 to 0.74 (Figure 1a)."

Comment B11:

The legend for Figure 1 and its caption contradict each other with respect to which significance level is shown.

[Figure]

Response:

Thank you for pointing out this important aspect. We have modified the legend accordingly.

Comment B12:

P7 L30: Maybe it would help to state your criteria earlier.

Response:

Accordingly, we have added the following sentence in the methodology.

"Specifically, the criteria we adopt for validation of the historical model simulations are, the ability of the models to reproduce the observed trends in surface temperature and rainfall over India during the summer monsoon season, and ability to simulate the observed negative correlation between the ISMR and the concurrent NINO3.4 Index."

Comment B13:

P8 L5: I think the formulation "of the corresponding statistic" is ambiguous. Maybe try to clarify.

Response:

The sentence was an inadvertent leftover from the multi-model mean statistics; as the reviewer had advised on an earlier version, we have removed most of the relevant text. We rewrite the relevant sentence as "The standard deviations of surface temperature, rainfall and NINO3.4 index are more or less comparable across the models, except for the $\sigma$ of the simulated NINO3.4 index from the FGOALS-s2 model, which is relatively higher."

Comment B14:

P8 L10: I'd like to note again as for the previous version: If I see it correctly, you use the GISS data uncorrected (compare https://www.clim-past-discuss.net/8/C393/2012/cpd-

8-C393-2012.pdf), I don't think that's too much of a problem, but you may note this somewhere. Sorry, that I didn't note this in my first review.

Response:

Thank you for the comment. We have rechecked the data for clarity. As we had already mentioned in the section 2, we have generated an ensemble mean of the above data, after filling in the missing values trough CDO.

Comment B15:

Paragraph: P8 L17ff: Calculating anomalies is a standard procedure. You don't tease out the signal with it. You just visualize the evolution more clearly. The reference to seasonal prediction is unnecessary. If there are outliers, then there are two outliers viewed globally. However, I don't think you validly can call them outliers especially considering that the models' 20th century evolution are much closer to each other than the past millennium data suggest. I also wouldn't speak of bias here.

Response:

Accordingly, we replaced the text of the first sentence "To tease out the signal more clearly...." with "To visualise the evolution more clearly....". The sentence now reads as "To visualise the evolution more clearly, we calculated the 101-year running mean of temporal anomalies of the TG (presented in Figure 4c) and TI (presented in Figure 4d).". We also removed the text on seasonal prediction and bias.

Comment B16:

P8 L25: You do not show this agreement with (paleo-)observations.

Response

Thanks. We modified the text to reflect the general agreement with the relevant discussion from paleo-observations presented in the technical section 5 of Stoker et al., 2013.

Comment B17:

Paragraph P8 L29ff: The discussion of the climatology-temperatures are interesting, but don't serve a purpose. We learn something about the models, but not about your topic. Further, line 32, it is not "interestingly", but obviously that the anomalies align better - you removed the climatological mean from the series.

Response:

The purpose of the discussion about the climatology-temperature was to compare the relative spread among the models during LM over global and Indian regions. Nonetheless, as indicated by the reviewer, we remove the text "Figures 2a and 2b indicate that the global mean temperature varies roughly 13°C to 16°C across the models through the LM. The corresponding range for the Indian subcontinent is 25°C to 29°C".

Further, in the erstwhile line 32, "interestingly" has been replaced by "Obviously", in light of the above comment.

Comment B18:

P9 L1: We know which volcanic eruption it likely was. See https://doi.org/10.1073/pnas.1307520110, https://www.nature.com/articles/srep34868, doi:10.1038/ngeo2875. We also know that other volcanic eruptions resulted in potentially decades long cooling episodes. One example of a potential reference is doi:10.1038/nature14565.

Response:

We cited the name of the volcanic eruption and added a sentence mentioning the decade long cooling effect of volcanic eruptions, all with relevant references, including the above.

Comment B19:

P9 L6: Well, some agreement. Are there further paleo-observations to show the agreement?

Response:

Thank you for the comment. The dataset we plotted is the most relevant, and publicly available. However, we have also cited all the available paleo-observational studies appropriately.

Comment B20:

Figure A3: What is plotted here? Global or Indian temperature? Is panel (a) just copied from the original publication or did you produce the Figure yourself. It looks strange. If you just adopt it, the publisher probably has to check the copyright?

Response:

As we have plotted the data in Fig. A3b after inferring the data from the original figure by using Digitizing software <http://getdata-graph-digitizer.com/>. Therefore, as you have indicated, we now remove the reproduction of the Figure A3a.

Comment B21:

P9 L12: Do you mean "MPI shows an insignificant weak decreasing trend"?

Response:

Yes, we do.

Comment B22:

P9 L15: I wouldn't speak of a limitation in this context. A spread is expected, the difference in signal between different periods may be seen as a limitation.

Response:

We removed the statement.

Comment B23:

P9 L22: I only count 5 simulations.

Response:

Table A3 clearly shows four such models, as mentioned in the sentence next to P9L22 of the earlier version. To avoid confusion, we removed the first sentence of the paragraph.

Comment B24:

P9 L24: I don't think the comparison of absolute values between different time-periods between historical observations and the simulations is valid. If you also provided the difference between the historical simulations and the historical observations, one could evaluate it against the general precipitation bias.

Response:

Thank you. We have verified and added relevant text on historical simulations and differences from the corresponding observations.

Comment B25:

P10 L7ff: I wonder under which circumstances the Indian subcontinent could have a colder absolute mean temperature than the globe as a whole. That is, I still don't see that this paragraph tells us anything of relevance.

Response:

We removed the text.

Comment B26:

Paragraph: P10 L11: If the standard deviations didn't change much, why is there a need to discuss them?

Response:

In our humble opinion, the finding, that the simulated standard deviations did not change much, is a result important enough to be documented at least briefly, as we did.

Comment B27:

P10 L29ff: If I understand you correctly, you're saying that the bootstrapping shows that the difference between correlation coefficients is significant in a number of models. Are you saying that the 0.05 in Figure A7 is the significance level? If so, please highlight it more clearly. This is the interesting result of the bootstrapping. Maybe it's just the caption that is unclear.

Response:

Thank you. As we mentioned in the caption of Figure A7, 0.05 is the correlation value representing 0.01 significance level (i.e. 90% confidence level) from a2-tailed Student's t-test

Comment B28:

My major concern implies that I don't think the sentences P11 L10 to L16 are valid as long as they relate to all events.

Response:

Indeed, accordingly, the revised discussion now pertains to only strong ENSO events (with magnitude greater than one standard deviation).

Comment B29:

P11 L14ff: The sentence should qualify that this holds for some simulations. Anyway, I am not sure I can follow your argument or even what you are referring to. Let me summarise what I see from your table - if I counted correctly: In the MWP there are in all models more El Niño (EN) than La Niña (LN). In the LIA five models have more or equal number of LN than EN. Two have more EN than LN. The absolute difference in

numbers between both types of events is larger for six models in the MWP and for one model in the LIA.

Response:

Thank you for pointing this out. We have cleaned up the text for clarity. Now it reads as "Interestingly, a majority of the PMIP3 models in this study indicate more strong El Niños as compared to strong La Niñas during the MWP than those of during LIA(Table 4). On the other hand, the number of strong La Niñas are marginally more than that of strong El Niños in all models during LIA than those of during MWP. Further, majority of models consistently simulate more number of strong El Niños (La Niñas) in MWP (LIA) as compared to the number of strong El Niños (La Niñas) in LIA (MWP); this result is statistically significant at 0.05 level from a 2-tailed Student's t-test carried out for difference of means."

Comment B30:

P11 L16ff: Didn't you state the BCC result in line 12 already?

Response:

We removed the statement.

Comment B31:

P11 L22: I am not sure whether the comparison between long term means and inter-annual variability is helpful.

Response:

We agree and remove the relevant reference.

Comment B32:

P14 L16ff: Different resolutions and structures of the models imply that patterns of change are not exactly the same - especially for precipitation.

Response:

Accordingly, we have added a sentence "To some extent, this disagreement in the distribution of anomalous changes in precipitation may be attributed to different resolutions and the physics of the models." after the statements that discuss of this different rainfall patterns.

Comment B33:

P14 L26: Your comparison is to the Last Millennium, isn't it? Thus, you cannot diagnose change relative to the historical period from this Figure, can you?

Response:

Thank you. We corrected it accordingly.

Comment B34:

L14 L26ff: This equatorial Indian Ocean change is hard to identify. It may be helpful to elaborate what you are referring to.

Response:

We modified the statement as suggested. Now it reads as "An anomalous divergence center over India resulted in relatively lesser rainfall during the LIA compared to the both MWP and LM."

Comment B35:

P15 L3: A 0.2 level is rather uncommon.

Response:

Indeed, 0.2 significance level is uncommon. Therefore, we rewrite the sentence and the one following that, emphasizing the rather weak correlation, as "This is also evidenced by the positive correlations between the LSTG at 850 hPa, derived from the ERA-20CM skin temperature (Hersbach et al. 2015) datasets, with the ISMR for the period

1901-2005, but statistically significant only at 0.2 level from a 2-tailed Student's t-test (Figure 8a). To account for the better reanalysis quality, we repeat the analysis for the 1950-1981 period, and these correlations are significant at 0.1 confidence level (Figure 8a)."

Comment B36:

P15 L8: Which negative correlations are you talking off? You don't show these, do you?

Response:

The LSTG-ISMR has the negative correlation during JJAS season. We have not shown a figure or table in the manuscript for the sake of brevity. Therefore, we have added "Figure not shown" to the text appropriately. A figure (Figure R2) with the correlations is provided below for your perusal.

Comment B37:

P15 L8: You mention the JJAS season, but do you show anything about the LSTG-ISMR-relationship then?

Response:

Kindly see the above response.

Comment B38:

P16 L1: Well, there are more climate forcings than just the anthropogenic.

Response:

Modified it accordingly. Removed sentence related to the forcings. Now the modified text reads as "The global climate has experienced significant centennial climate variations in the last two millennia (IPCC, 2013)"

Comment B39:

P16 L14ff: This sentence is still not relevant.

Response:

We removed the sentence.

Comment B40:

P16 L20: What do you mean by: "statistically significant . . . in comparison to the current day climate"?

Response:

We modified the sentence.

Comment B41:

P16 L24ff: You find this modulation in some simulations.

Response:

For clarity, we merged this sentence and next one. The revision now reads as "Indeed, we find a multi-centennial modulation of the simulated ENSO-ISMR correlations; at least four models suggest a decreasing ENSO-ISMR (as well as that with the Indian summer temperatures)...."

Comment B42:

P16 L27: Does this refer to all events? If so, my major concern applies.

Response:

Modified the relevant text to talk about only strong events.

Comment B43:

P16 L28: Why is it "despite" the occurrence?

Response:

In the current period, El Niños are normally associated with a westward shift. Therefore, the use of "despite".

Comment B44:

P16 L28: Is this relatively high compared to the Last Millennium? Did you present this change relative to the full period?

Response:

Sorry for the typo. We modified the statement. It refers to LIA not LM.

Comment B45:

P17 L5: What do mean by "the spatial distribution . . . is . . . higher"?

Response:

For clarity, we have rewritten the sentence as follows "The simulated surface temperature over India is only modestly higher during the MWP as compared to the corresponding LM average, owing to the spread of the signals across the models."

Comment B46:

P17 L18: Does this relate to all events including neutral? See my major concern.

Response:

This relates to only strong events. We modified the text as suggested.

Technical Issues:

Comment C1

Please improve the quality of the Figures. For example, produce them in the correct aspect ratio and do not rescale them in a graphics software.

Response:

[Figure]

We improved the picture quality as suggested.

Comment C2

Please be consistent in using Figure or Table AN or SN, i.e., please check that you always use the abbreviation A for Appendix.

Response:

We have gone through it and corrected accordingly. They are now labelled properly.

Comment C3

There are a number of typos etc. but I leave those for the copy-editor. There are also further unclear formulations. Some of these appear to have survived from the last two revisions.

Response:

We have revisited the manuscript carefully and eliminated the typos.

Comment C4

P6 L30: Remove "We revise the text accordingly"

Response:

Sorry for this faux pas. Removed it.

Comment C5

P8 L25: I again think your reference is incorrectly formulated here. There are further instances of this.

Response:

Corrected it.

Comment C6

P11 L23: This sentence is already in the beginning of the paragraph.

Response:

We removed it.

Comment C7

P12 L14: Please check that this paragraph, the captions, and the content of Figure 5 and Table 5 are correct.

Response:

We made necessary modifications.

Comment C8

Paragraph P13 L23ff. Shouldn't this come before the Paragraph P13 L10ff?

Response:

Thank you. We modified it.

Comment C9

Figure A15: The second Figure A15 should be Figure A16, I guess.

Response:

Ok. Modified it accordingly.

Comment C10

P15 Footnote: This should be Figure A16.

Response:

Done.

Comment C11

P15 L27: I would write "such as volcanic eruptions" instead of "such as more volcanic eruptions during the LIA".

Response:

Done.

Comment C12

P17 all: You mix the results for Temperature and Precipitation in a way that may confuse the reader. Restructuring this section may help.

Response:

Thank you. We have now separated this text into two paragraphs. The first paragraph is mainly about the rainfall, as suggested. The second paragraph discusses temperature changes, and plausible reasons for the warm and wet conditions over India during MWP (e.g. Lehmann et al., 2015; Goswami et al., 2006).

Comment C13

P17 L18: seven out of seven should possibly read all?

Response:

Corrected it.

Figure R1: ISMR trend lines for Historical simulations by using Mann-Kendall test for CE 1980-2005.

Figure R2: Correlation between LSTG-ISMR during JJAS season for both MWP and LIA.

Please also note the supplement to this comment:
https://www.clim-past-discuss.net/cp-2018-7/cp-2018-7-AC1-supplement.pdf

[Figure]

[Figure]

**Fig. 1.** Figure R1: ISMR trend lines for Historical simulations by using Mann-Kendall test for CE 1980-2005.

[Figure]

**Fig. 2.** Figure R2: Correlation between LSTG-ISMR during JJAS season for both MWP and LIA.

| BCC | | CCSM4 | | GISS | | HADCM3 | | IPSL | | MPI | | FS2 | |
|---|---|---|---|---|---|---|---|---|---|---|---|---|---|
| El Niños | La Niñas | El Niños | La Niñas | El Niños | La Niñas | El Niños | La Niñas | El Niños | La Niñas | El Niños | La Niñas | El Niños | La Niñas |
| 12 | 9 | 12 | 5 | 13 | 10 | 5 | 2 | 14 | 7 | 10 | 6 | 16 | 7 |

**Fig. 3.** Table S1 Number of Strong (only events whose amplitude is above $1\sigma$) ENSO events after CE 1950.

---

## Author Comment (AC2) · 11 May 2018

Summary:

The authors use 7 PMIP3 simulations to investigate the ENSO teleconnection to the Indian Summer Monsoon during the Last Millennium. The Author use the present day period to evaluate the model simulations and compare their results to some of the existing proxy reconstructions. The authors claim that during the Medieval Climate Anomaly the frequency of El Nino events is enhanced whereas during the Little Ice Age La Nina events occur more frequently. Then, they discuss some non-linearity which is unfortunately not presented in an understandable way.

Overall judgment

I see that the authors put a lot of effort to analyze PMIP3 simulations. However, the way how the results are presented and even more importantly the questionable content leads to the manuscript lacks a clear structure; the phrasing is inadequate, preventing the reader to understand the content. Furthermore, the main result of the study, that during the Medieval Climate Anomaly (MCA) the frequency of El Nino events is enhanced whereas during the Little Ice Age (LIA) La Nina events occur more frequently is questionable as detailed below. So, I recommend to reject the manuscript from publication in Climate of the Past.

Response:

Dear Reviewer, we are grateful for noticing our efforts. The manuscript has gone through several revisions in light of comments from DR. Bothe, the first reviewer, and DR. Wenmin Man (who reviewed an earlier version, and the editor. We greatly appreciate your time in perusing our manuscript, and for the constructive comments. Considering these, and those from reviewer and the Editor, we will carefully revise the manuscript. In fact, as the comments from the reviewer 1, DR. Bothe, arrived a few weeks earlier, we have carefully revised the manuscript, among other things, for better clarity and understanding as also suggested by him. We shall also incorporate the suggestions from you soon, and now hopefully, the revised version would meet your requirements, and the standard of the Climates of the Past.

General Comments

Comment GC1:

I. The manuscript needs a serious proof reading by a native speaker.

Response:

Thank you. This suggestion will be carefully implemented.

Comment GC2:

II. The structure of the manuscript is not clear, e. g., section 3.4 contains again an evaluation part. Presenting 'preliminary results' in a manuscript makes no sense, either the results are solid and necessary or not (then they shall not be presented). The authors made no clear selection of figures. It looks like the 'randomly' selected eight figures (+5 tables) in the main part and put the rest of the analysis made in the appendix (which is 15 figures and 4 tables).

Response:

We are sorry for the confusion, which arose only due to the wrongly-formed title for the subsection 3.4. The results reported in this subsection are not preliminary at all. The section 3.4 contains dynamics such as large-scale convergence/divergence patterns and exploration of Land-Sea thermal gradient. To avoid confusion, we remove the words "preliminary analysis" from the title of sub-section 3.4.

We have already received a similar suggestion from DR. Bothe. Accordingly, we plan to move the relevant figures into the main text, and the supplemental information now only contains only 6 Figures and 5 Tables.

Comment GC3:

III. The manuscript builds on one main finding, namely an increase of El Nino events during MCA and an increase of La Nina events during LIA. The authors ignore the fact that they use the NINO3.4 index which by definition varies a bit with the global mean signal. Thus, if the global mean temperature due to external forcing is increase the Nino3.4 index will certainly be biased positive and lead to or El Ninos (although the cause is a global signal and not a real change in ENSO). The authors already show in their results that ENSO is NOT changing from the MCA to the LIA as the standard deviation during the periods is the same (see page 10).

Response:

Thank you. we find that a majority of the PMIP3 models in this study indicate more El Niños as compared to the La Niñas during the MWP (and relatively less number less number of El Niños as compared to the La Niñas during LIA). This is notwithstanding the relatively unchanging standard deviation of the NINO3.4 index across the LIA & MWP (as shown Tables A2 and A4of the submitted manuscript for discussion round).

We agree that the mean background changes in temperatures may modulate the relative strengths of El Niños and La Niñas (e.g. Federov & Philander 2000), thereby introducing a non-linearity in the relative strengths/frequencies of the warm & cold ENSO phases. In the revision, we shall mention this aspect as a possible cause for more El Niños in MWP.

Comment GC4:

IV. All figures are of bad quality.

Response:

We have improved the pictures clarity.

Technical comments

Comment TC1:

L20-21: Unclear sentence

Response

Revised the sentence for clarity.

Comment TC2:

L22-28: Awkward and unclear statements.

Response:

We modified the statements. Modified statements now it reads as "Interestingly, the percentage of the simulated strong El Niños associated with negative ISMR anomalies is higher in the LIA. Also, the percentage of strong La Niñas associated with positive ISMR anomalies is higher in the MWP. This non-linearity is apparently important for the relatively higher ISMR during the MWP. Further, distribution of simulated anomalous boreal summer velocity potential at 850 hPa during MWP in models indicates a zone of anomalous convergence in the central tropical Pacific flanked by two zones of divergence. This suggests a westward shift in the Walker circulation as compared to the mean pattern of the 850 hPa convergence and divergence. The simulated 850 hPa walker circulation during the MWP is also prominent relative to the corresponding historical simulations."

Comment TC3:

L28: divergence center of what??

Response:

Divergence center of anomalous 850 hPa circulation calculated using the zonal and meridional winds

Comment TC4:

L29: convergence of what??

Response:

Convergence center of anomalous 850 hPa circulation calculated using the zonal and meridional winds

Comment TC5:

L30: Connection between the two parts separated by a semicolon is not given.

Response:

Thanks. A semi-colon has been replaced by a period.

Page 2:

Comment TC6:

L6: IPCC (2013) is not an adequate reference here, please use more specific references

Response:

Thank you. In addition to the IPCC, we also cite the PAGES 2k Consortium (2013).

Comment TC7:

L7-9: Unclear statement

Response:

We modified the statement.

Comment TC8:

L11-15: Missing references of definition of time periods of MWP and LIA also for the variation of the periods you need to give references.

Response:

We modified the text. Now it reads as "Paleo-data based studies identify two significant periods in the last millennium (LM), i.e. Common Era (CE) 0850-1849, prior to when the instrumental observations started. These two periods are, (i) a relatively warmer period known in literature as the 'Medieval Warm Period' (MWP, CE 950-1350), roughly followed by (ii) a relatively cooler period, the Little Ice Age (LIA, CE 1500-1850) (e.g. Lamb et al, 1965; Grove et al, 1988; Graham et al, 2010; Mann et al, 2009)."

Comment TC9:

L18: Be more specific about the regions you are referring to.

Response:

Paleoclimate reconstructions from various well-dated proxy data suggest that during the MWP, some regions experienced temperatures as warm as mid-20th century, whereas some others were as warm as the late-20th century (e.g. extratropics, southern hemisphere land region; Stocket el al., 2013).

Comment TC10:

Paragraph 3: There is no logical connection to the paragraph before

Response:

Thank you. The paragraph now reads as "Paleoclimate reconstructions from various well-dated proxy data suggest that during the MWP, some regions experienced temperatures as warm as mid-20th century whereas some others were as warm as the late-20th century (e.g., IPCC 2013, Fleitmann et al., 2007; Borgaonkar et al., 2010; Ponton et al., 2012). As can be seen, these studies do not report the conditions at a regional scale. Particularly, there are no proxy or modelling studies that have reported on the temperature conditions over the Indian subcontinent, which is a major hotspot of climate variability, largely from the perspective of the summer monsoon rainfall."

Now this paragraph connects logically to the next one, which reads as

"The Indian Summer Monsoon Rainfall (ISMR; June-September; JJAS) variability is manifested on intra-annual, interannual, decadal, centennial and millennial to multi-millennial time scales (Ramesh et al., 2010)......."

Comment TC11:

L33-34: Awkward sentences, please clarify.

Response:

We modified the sentence for better clarity. Now it reads as "Proxy records also suggest

that during the last millennium, ISMR was the higher during the MWP and relatively weaker during the LIA (Yadava et al., 2005)."

I stop here as the entire manuscript is like the first two pages.

These issues have been carefully addressed, thanks to Dr. Bothe, the reviewer 1, who had also kindly given many such suggestions on the complete manuscript.

Please also note the supplement to this comment:
https://www.clim-past-discuss.net/cp-2018-7/cp-2018-7-AC2-supplement.pdf